# Inhibition of IRE1 RNase activity modulates the tumor cell secretome and enhances response to chemotherapy

Susan E. Logue [1,2], Eoghan P. McGrath[1,2], Patricia Cleary[1,2], Stephanie Greene[3], Katarzyna Mnich [1,2], Aitor Almanza[1,2], Eric Chevet[4,5], Róisín M. Dwyer [6], Anup Oommen [2], Patrick Legembre[4,5], Florence Godey[4,5], Emma C. Madden [1,2], Brian Leuzzi[1,2], Joanna Obacz[4,5], Qingping Zeng [7], John B. Patterson[3], Richard Jäger [8], Adrienne M. Gorman [1,2] & Afshin Samali [1,2]

Triple-negative breast cancer (TNBC) lacks targeted therapies and has a worse prognosis than other breast cancer subtypes, underscoring an urgent need for new therapeutic targets and strategies. IRE1 is an endoplasmic reticulum (ER) stress sensor, whose activation is predominantly linked to the resolution of ER stress and, in the case of severe stress, to cell death. Here we demonstrate that constitutive IRE1 RNase activity contributes to basal production of pro-tumorigenic factors IL-6, IL-8, CXCL1, GM-CSF, and TGFβ2 in TNBC cells. We further show that the chemotherapeutic drug, paclitaxel, enhances IRE1 RNase activity and this contributes to paclitaxel-mediated expansion of tumor-initiating cells. In a xenograft mouse model of TNBC, inhibition of IRE1 RNase activity increases paclitaxel-mediated tumor suppression and delays tumor relapse post therapy. We therefore conclude that inclusion of IRE1 RNase inhibition in therapeutic strategies can enhance the effectiveness of current chemotherapeutics.

[1] Apoptosis Research Centre, National University of Ireland Galway, University Road, Galway H91 TK33, Ireland. [2] School of Natural Sciences, National University of Ireland Galway, University Road, Galway H91 TK33, Ireland. [3] Fosun Orinove PharmaTech Inc., 3537 Old Conejo Road, Suite 104, Newbury Park, CA 91320, USA. [4] Inserm U1242, Chemistry Oncogenesis Stress Signaling, Université de Rennes 1, Avenue de la Bataille Flandres Dunkerque, 35042 Rennes, France. [5] Centre de Lutte Contre le Cancer Eugène Marquis, Avenue de la Bataille Flandres Dunkerque, 35042 Rennes, France. [6] LAM 2015, Discipline of Surgery, Lambe Institute for Translational Research, National University of Ireland Galway, Costello Road, Galway H91 V4AY, Ireland. [7] Fosun Orinove PharmaTech Inc., Suite 211, Building A4, 218 Xinghu St., Suzhou Industrial Park, Jiangsu 215123, China. [8] Hochschule Bonn-Rhein-Sieg, University of Applied Sciences, Department of Natural Sciences, von-Liebig-Straße 20, 53359 Rheinbach, Germany. Correspondence and requests for materials should be addressed to A.S. (email: afshin.samali@nuigalway.ie)

nositol requiring enzyme 1 alpha (referred to as IRE1 hereafter, also known as ERN1), an endoplasmic reticulum (ER) resident type I transmembrane protein, is composed of an N-terminal ER luminal domain and a C-terminal cytosolic domain that possesses both kinase and endoribonuclease (RNase) activities. IRE1 function has been studied extensively during ER stress where it constitutes an important pro-survival arm of the unfolded protein response (UPR)[1]. Accumulation of unfolded proteins in the ER (ER stress) triggers IRE1 dimerization and trans-autophosphorylation facilitating its activation[2]. Activated IRE1 cleaves X-Box Binding Protein 1 *(XBP1)* mRNA via its RNase activity[3]. Subsequent re-ligation of *XBP1* mRNA, by RNA 2′,3′-cyclic phosphate and 5′-OH ligase (RTCB), permits translation of a transcription factor referred to as spliced XBP1 (XBP1s)[4]. XBP1s has predominantly been studied within the context of the UPR where its target genes encode mainly adaptive, pro-survival factors involved in ER homeostasis[5]. However, recent studies indicate that XBP1s has a much broader range of target genes than previously appreciated. For example, selective ablation of IRE1/XBP1s signaling in lipopolysaccharide (LPS)-treated macrophages reduced interleukin (IL)-6 and IL-8 production, thus attenuating pro-inflammatory responses[6]. In addition to XBP1 splicing, IRE1 RNase activity facilitates selective degradation of RNA by directly cleaving cytosolic RNA species, in a process referred to as regulated IRE1 dependent decay (RIDD)[7]. Similar to the IRE1–XBP1s axis, RIDD signaling has been predominantly examined in cellular stress responses where it is associated with both pro-survival and pro-death roles depending upon the duration and severity of the initiating stress[8,9].

The UPR, and in particular, the IRE1–XBP1 branch, has been linked to tumor development, progression, and post-therapy responses in a wide range of cancers including breast, prostate, and pancreatic cancer[10–13]. The precise mechanism by which IRE1 RNase signaling promotes cancer progression in these settings is not fully understood. Nevertheless, the IRE1–XBP1s signaling axis has emerged as a potential therapeutic target in cancer leading to the development of small molecule inhibitors targeting the IRE1 RNase domain[14–17]. However, the majority of current IRE1 RNase inhibitors have poor pharmacodynamic properties rendering their use as clinical agents unlikely.

In this study, we evaluate the outcome of blocking IRE1 RNase activity in triple-negative breast cancer (TNBC) cells using a small molecule inhibitor—MKC8866. MKC8866 is a selective IRE1 RNase inhibitor that exhibits acceptable pharmacokinetic and toxicity profiles, making it an attractive agent for pre-clinical development. Inhibition of IRE1 RNase activity by MKC8866 in breast cancer cells leads to the decreased production of pro-tumorigenic factors including IL-6, IL-8, chemokine (C-X-C) ligand 1 (CXCL1), transforming growth factor β 2 (TGFβ2), and granulocyte-macrophage-colony-stimulating-factor (GM-CSF), linking constitutive IRE1 RNase activity to maintenance of a pro-tumorigenic secretome.

Chemotherapy-induced modulation of the secretome is a known promoter of tumor relapse[18,19]. Paclitaxel, a commonly used chemotherapeutic for the treatment of TNBC, has been linked to the production of pro-tumorigenic factors[18,19]. Our results demonstrate that this occurs in a manner partly dependent on IRE1 RNase activity, leading us to propose that the combination of IRE1 RNase inhibitors with chemotherapeutics, such as paclitaxel, may be more efficacious than chemotherapy alone. Indeed, we observe decreased mammosphere formation post-paclitaxel treatment in MKC8866-treated TNBC cells compared to those treated with vehicle alone. Likewise, in vivo, MKC8866 administered in combination with paclitaxel enhances the effectiveness of paclitaxel and limits tumor regrowth upon cessation of paclitaxel treatment.

## Results

**Breast cancer cells exhibit constitutive IRE1 RNase activity.** A panel of breast cancer cell lines encompassing the main molecular subtypes (estrogen receptor positive—MCF7, T47D, Human Epidermal growth factor Receptor 2 (HER2) positive—SKBR3 and triple negative—MDA-MB-231, MDA-MB-468) was examined for basal IRE1 RNase activity by assessing levels of spliced XBP1. In all breast cancer lines tested, *XBP1s* mRNA was detected, to varying degrees, with the highest levels present in TNBC cells (Fig. 1a). Examination of XBP1s protein expression revealed a similar pattern with the highest expression evident in the TNBC cell lines MDA-MB-231 and MDA-MB-468 (Fig. 1b). MCF10A, a spontaneously immortalized, non-transformed, non-tumorigenic breast epithelial cell line, did not display basal IRE1 RNase activity (Fig. 1a, b). However, treatment of MCF10A cells with the ER stress inducer, Tunicamycin (Tm), triggered significant XBP1 splicing, indicating that IRE1 RNase, while not constitutively active, is functional in these cells (Fig. 1a, b). In addition to commonly used breast cancer cell lines IRE1 RNase activity was also assessed by quantitative PCR (Q-PCR) in a range of primary patient samples. Similar to the results obtained in the breast cancer cell lines, the ratio of spliced to total *XBP1* was highest in samples derived from basal-like breast cancers (most of which are TNBC) compared to luminal samples, and tumor-associated normal tissue (TAN) (Fig. 1c).

**Inhibition of IRE1 reduces breast cancer cell proliferation.** MKC8866 (Fig. 2a) is a member of a small molecule IRE1 RNase inhibitor family first described by Patterson and colleagues in 2011[14]. It is a salicylaldehyde analog, that binds to IRE1 within the RNase catalytic site and inhibits both XBP1 splicing and RIDD activity[14]. Addition of MKC8866 rapidly attenuated basal IRE1 RNase activity in MDA-MB-231 cells, as demonstrated by decreased levels of *XBP1s* transcript and its downstream targets endoplasmic reticulum DNA J domain-containing protein 4 *(ERDJ4,* also known as *DNAJB9)* and homocysteine-responsive ER protein with ubiquitin like domain 1 *(HERP,* also known as *HERPUD1)* (Fig. 2b). Addition of MKC8866 blocked Tm-induced IRE1-mediated signaling but did not affect Tm-induced activation of the other two arms of the UPR, protein kinase R (PKR)-like endoplasmic reticulum kinase (PERK) or activating transcription factor 6 (ATF6) (Fig. 2c). Indeed, neither PERK phosphorylation and downstream CHOP induction, nor ATF6 processing were affected by MKC8866, underscoring the selectivity of MKC8866 for IRE1 under both basal and stress conditions. To examine the effect of the constitutive IRE1 signaling observed in breast cancer cells, IRE1 RNase activity was blocked by addition of MKC8866 and the outcome on cell proliferation/viability was assessed (Fig. 2d, e, Supplementary Fig. 1a, b). Addition of MKC8866 decreased proliferation of all breast cancer cell lines tested without inducing cell death (Fig. 2d, Supplementary Fig. 1a). Cell cycle analysis, using 5-ethynyl-2′-deoxyuridine (EdU) incorporation, indicated that inhibition of IRE1 RNase activity by MKC8866 reduced the number of cells entering S phase (Supplementary Fig. 1b). In contrast, addition of MKC8866 to MCF10A cells, which do not display constitutive IRE1 RNase activity, did not alter cell proliferation (Fig. 2d, Supplementary Fig. 1c). Knockdown of XBP1 or IRE1 in MDA-MB-231 cells similarly reduced cell proliferation when compared to controls (Fig. 2f, g, Supplementary Fig. 1d-g).

**IRE1 gene signature associates with basal-like breast cancer.** Transcriptomic data obtained from MKC8866 versus vehicle-only microarray experiments identified 401 differentially expressed probe-sets representing 395 genes. These initial candidate

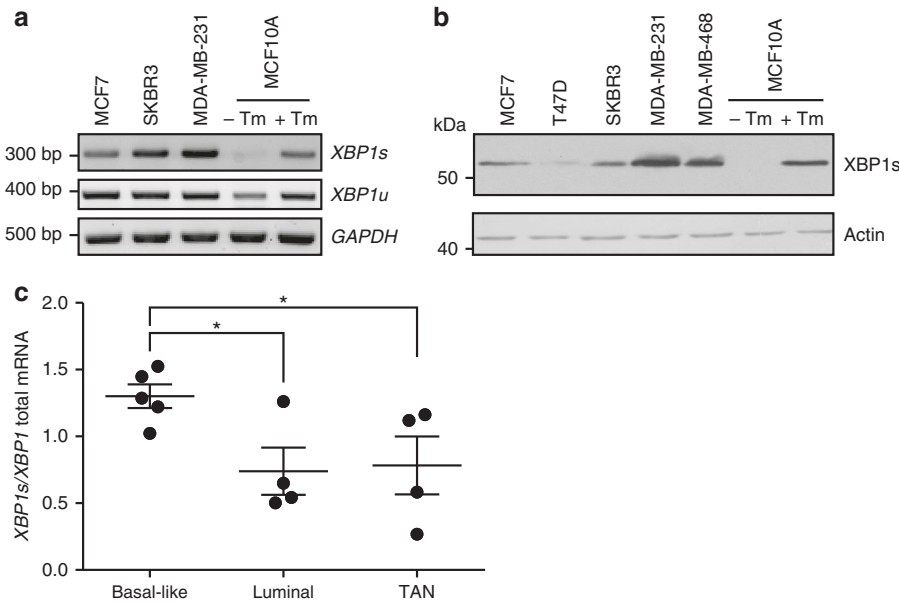

**Fig. 1** Breast cancer cells exhibit constitutive IRE1 RNase activity. **a** Expression of spliced and unspliced *XBP1* mRNA was assessed in a panel of breast cancer cell lines (MCF7, SKBR3, MDA-MB-231) and the non-tumorigenic breast epithelial cell line MCF10A via RT-PCR. Tunicamycin (Tm) (1 μg ml$^{-1}$)-treated MCF10A cells act as a positive control for *XBP1* splicing. *GAPDH* was used as a loading control. **b** Immunoblotting of XBP1s in a panel of breast cancer cell lines (MCF7, T47D, SKBR3, MDA-MB-231, and MDA-MB-468) and the non-tumorigenic breast epithelial cell line MCF10A. Tm (1 μg ml$^{-1}$)-treated MCF10A cells act as a positive control for XBP1 splicing. Actin was used as a loading control. **c** Q-PCR quantification of the relative mRNA levels of spliced to total *XBP1* in RNA samples obtained from basal-like tumor tissue ($n = 5$), luminal tumor tissue ($n = 4$), and tumor-associated normal (TAN) tissue ($n = 4$). Results shown for **a** and **b** are representative of three independent experiments. *$P < 0.05$ based on a pairwise single factor ANOVA tests comparing each tissue type. Error bars represent s.e.m.

markers were tested in a training dataset containing gene expression profiles of a panel of 27 breast cancer cell lines. Genes with a strong positive or negative correlation with IRE1 activity were further prioritized using BioInfoMiner to generate an 83-gene signature predictive of IRE1 activity (IRE1 gene signature) (Supplementary Table 1). This IRE1 gene signature was applied to the 27 breast cancer cell lines, ranking them based on their predicted level of IRE1 RNase activity (Fig. 3a). The resultant ranking largely mirrored results achieved by western blotting for XBP1s (Fig. 1b) with those cell lines representative of TNBC being ranked highest (MDA-MB-231) while those lines representative of HER2-positive (SKBR3) and estrogen receptor-positive (T47D) ranked lower. We then applied the IRE1 gene signature to a cohort of 595 breast cancer tumors from The Cancer Genome Atlas (TCGA) database and, using the same approach, identified two distinct subsets of patients indicative of low ($n = 79$) and high IRE1 ($n = 63$) activity (Fig. 3b). Analysis of the breast cancer subtype in each, based on PAM50 classification criteria[20], revealed that a high IRE1 gene signature associated exclusively with basal-like breast cancers, while tumors associated with a low IRE1 gene signature were predominantly of the luminal subtype (Fig. 3c). Gene set enrichment analysis (GSEA) indicated cancers with an elevated IRE1 gene signature associated with a more mesenchymal-like phenotype, increased invasiveness and a worse clinical outcome (Supplementary Fig. 2).

**IRE1 activity induces production of pro-inflammatory factors.** To identify biological processes associated with constitutive IRE1 RNase activity in breast cancer patients, we again applied the IRE1 gene signature to the 595 breast cancer tumors from TCGA database. An IRE activity score for each tumor was predicted based on expression levels of the 83 genes from the IRE1 gene signature and the Pearson correlation coefficient was calculated between the score and every other gene across 595 tumors to

produce a ranked gene list. We then performed GSEA using gene ontology terms on the ranked gene list. Using this approach, we found that predicted IRE1 activity strongly associates with the expression of genes involved in inflammatory responses (Supplementary Fig. 3). We also noted that genes encoding pro-inflammatory factors (*IL6*, *IL8*, and *TGFB2*) comprised a subset of the downregulated genes in the microarray experiment, suggesting a link between IRE1 RNase activity and the production of pro-inflammatory factors in TNBC cells.

Comparison of *IL6*, *IL8*, *GM-CSF*, *CXCL1*, and *TGFB2* mRNA expression levels between the IRE1 RNase high and IRE1 RNase low activity populations within the 595 patient cohort from TCGA revealed a significantly higher expression level of each pro-inflammatory factor in the IRE1 RNase high grouping (Fig. 3d). Examination of pro-tumorigenic cytokine and XBP1s expression, via immunohistochemistry, showed that high XBP1s expression positively correlates with elevated IL-8 and CXCL1 staining in human TNBC tissue sections (Fig. 3e, f) supporting our in silico findings.

To further investigate this link we generated conditioned medium from MDA-MB-231 cells following 48 h treatment with MKC8866 or vehicle alone. Once equal cell number post treatment was confirmed (Supplementary Fig. 4a), the conditioned medium was applied to a cytokine array assaying 102 different factors. Using this approach we observed reduced levels of IL-6, IL-8, GM-CSF, and CXCL1 in MKC8866 conditioned medium compared to vehicle-only conditioned medium (TGFβ2 was not present on the array) (Fig. 4a). Using a combination of ELISAs and Q-PCR, we confirmed that inhibition of IRE1 RNase activity by MKC8866 treatment reduced the production and secretion of IL-6, IL-8, CXCL1, GM-CSF, and TGFβ2 in MDA-MB-231 cells (Fig. 4b–d). We also tested the effect of MKC8866 addition on IL-6, IL-8, CXCL1, GM-CSF, and TGFβ2 secretion in three additional TNBC cell lines (MDA-MB-468, BT-549, and

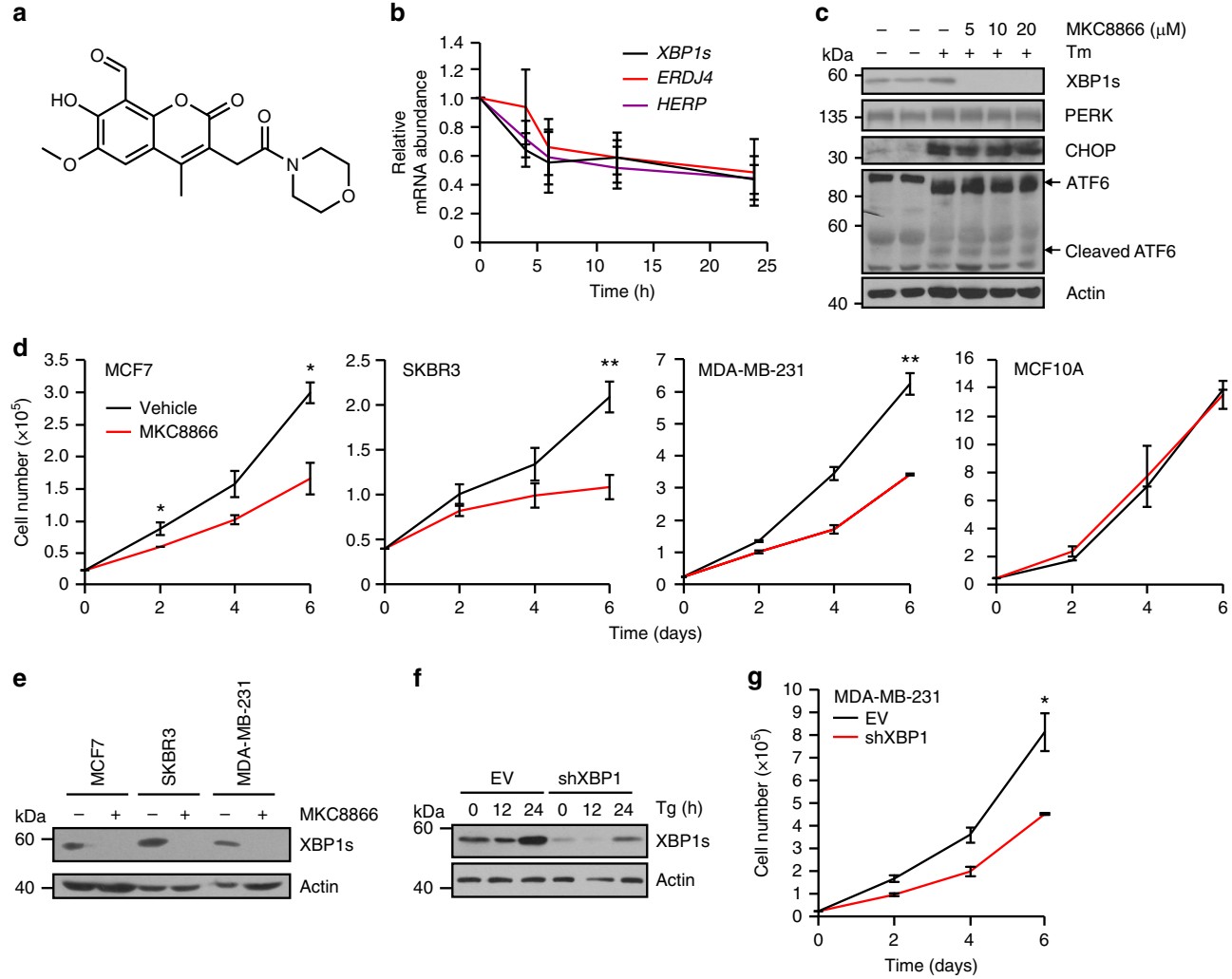

**Fig. 2** Inhibition of IRE1 reduces breast cancer cell proliferation. **a** Chemical structure of MKC8866. **b** MDA-MB-231 cells were treated with 20 μM MKC8866 for 4, 6, 12, and 24 h after which RNA was extracted and levels of *XBP1s*, *ERDJ4*, and *HERP* mRNA transcripts quantified by Q-PCR (n = 4). **c** T47D cells were treated for 24 h with 1 μg ml⁻¹ Tm alone or in combination with increasing concentrations of MKC8866 (5, 10, 20 μM) and cell lysates immunoblotted for XBP1s, PERK, CHOP, ATF6, and Actin. **d** MCF7 (n = 3), SKBR3 (n = 5), MDA-MB-231 (n = 2), and MCF10A (n = 3) cells were treated with 20 μM MKC8866 or an equal volume of DMSO and cell proliferation monitored by cell counts every second day for 6 days. **e** MCF7, SKBR3, and MDA-MB-231 cells were treated with 20 μM MKC8866 for 24 h and cell lysates immunoblotted for XBP1s and Actin. **f** Empty vector (EV) and XBP1shRNA MDA-MB-231 cells were treated for the indicated times with Tg (0.5 μM) after which expression of XBP1s and Actin was determined by immunoblotting. **g** Proliferation of empty vector (EV) and XBP1shRNA MDA-MB-231 cells was monitored by cell counts every second day for 6 days (n = 3). Results shown for **c**, **e**, and **f** are representative of three independent experiments. *P < 0.05, **P < 0.01 and ***P < 0.001, based on a Student's t test. Error bars represent s.e.m.

HCC1806). However, CXCL1 was the only common factor significantly reduced by MKC8866 (Fig. 4e) in all cell lines tested, suggesting IRE1 RNase activity may be of particular importance in regulating CXCL1 production in TNBC cells.

To validate that the effect of MKC8866 on cytokine production was indeed a consequence of reduced IRE1 RNase signaling we knocked down IRE1 expression in MDA-MB-231 cells by siRNA and examined secretion of IL-6, IL-8, TGFβ2, GM-CSF, and CXCL1. With the exception of TGFβ2, IRE1 knockdown reduced the secretion of each cytokine to levels observed in non-targeting siRNA controls (NC) treated with MKC8866 (Supplementary Fig. 4b-d). In agreement with these data, analysis of transcript levels demonstrated a similar pattern with a reduction in all factors, again with the exception of TGFβ2, observed in IRE1 knockdown cells compared to NC (Supplementary Fig. 4e). Likewise, knockdown of XBP1 reduced transcript levels of *IL6*, *IL8*, *GM-CSF*, *CXCL1*, and to a lesser extent *TGFB2*

(Supplementary Fig. 4e). Since TGFβ2 regulation upon IRE1 knockdown differed to the results obtained with MKC8866, we questioned whether this was an off-target effect of MKC8866. To examine this, we added MKC8866 to both NC and IRE1 knockdown cells. While MKC8866 suppressed TGFβ2 secretion in NC controls it failed to do so in IRE1 knockdown cells verifying its reliance on IRE1 expression (Supplementary Fig. 4d). It is possible that reducing protein expression of IRE1 triggers a compensatory increase in TGFβ2 production through IRE1-independent mechanisms.

Collectively, these findings indicate that constitutive IRE1 RNase activity in MDA-MB-231 cells contributes to the composition of the secretome and that addition of a small molecule IRE1 RNase inhibitor, MKC8866, can limit the production of secreted pro-tumorigenic factors. Since MKC8866 addition slowed the proliferation rate of MDA-MB-231 cells we asked whether any of the cytokines identified as being regulated

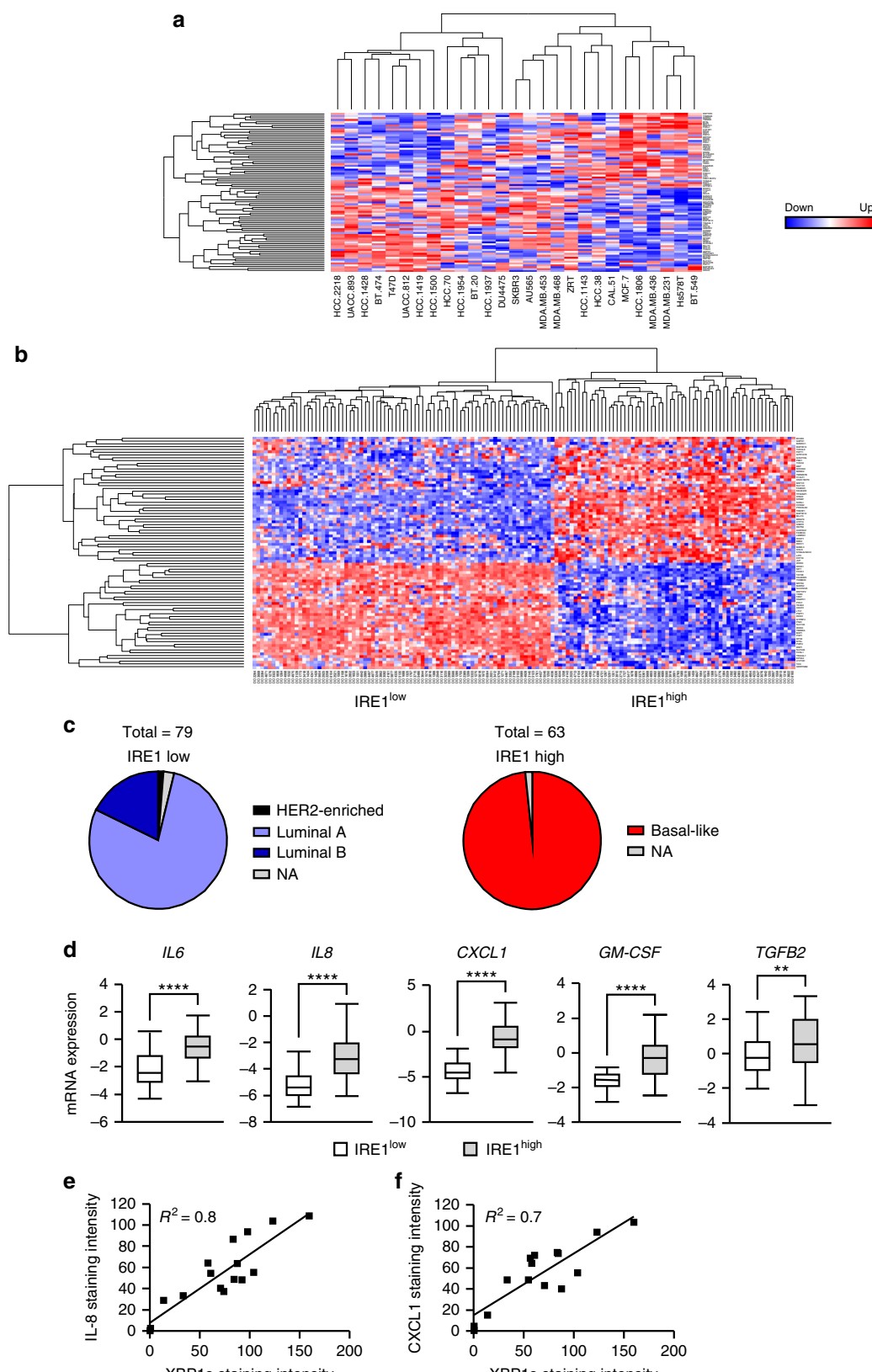

in an IRE1 RNase-dependent manner contributed to cell proliferation. To this end, we incubated MDA-MB-231 cells with either MKC8866 or neutralizing antibodies against each of CXCL1, IL-6, IL-8, GM-CSF, and TGFβ2. We observed a decrease in MDA-MB-231 cell proliferation, comparable to that observed with MKC8866, following administration of anti-

CXCL1, IL-6, IL-8, GM-CSF, and TGFβ2 antibodies (Supplementary Fig. 4f).

**Paclitaxel increases IRE1-dependent cytokine secretion.** Neo-adjuvant chemotherapy is the current standard of care for TNBC

**Fig. 3** IRE1 gene signature associates with basal-like breast cancers. **a** The putative IRE1 RNase-dependent gene signature was applied to a gene expression data set comprised of 27 breast cancer cell lines. Hierarchical clustering was performed and cell lines ranked based on their predicted IRE1 RNase activity. Expression across each gene (row) was centered and scaled so that mean expression is zero and standard deviation is one. Red indicates those genes with high expression and blue those with low expression relative to the mean. **b** IRE1 RNase-dependent gene signature was used to stratify 595 breast cancer gene expression data sets in TGCA. Cohorts with high and low IRE1 activity where identified and are represented as a heat map. Red indicates genes with high expression while blue those with low expression relative to the mean. **c** Pie charts depicting the breast cancer molecular sub-types (based on PAM50 classification) of IRE1 high and IRE1 low cohorts. NA indicates samples where PAM50 classification information was not available. **d** mRNA expression levels of *IL6, IL8, CXCL1, GM-CSF*, and *TGFB2* in IRE1 high versus IRE1 low cohorts. Box plots show the median, 25th and 75th percentiles, and whiskers indicate the location of the minimum and maximum values for each of the IRE1 low ($n = 79$) and IRE1 high ($n = 63$) groups. **e** Correlation between immunohistochemistry staining intensity for XBP1s and IL-8 in TNBC human tumor sections ($n = 16$). **f** Correlation between immunohistochemistry staining intensity for XBP1s and CXCL1 in human TNBC tumor sections ($n = 14$). *$P < 0.05$, **$P < 0.01$, ***$P < 0.001$, and ****$P < 0.0001$, based on comparison of the two groups using a two-tailed $t$ test with Welch's correction. Error bars represent s.e.m.

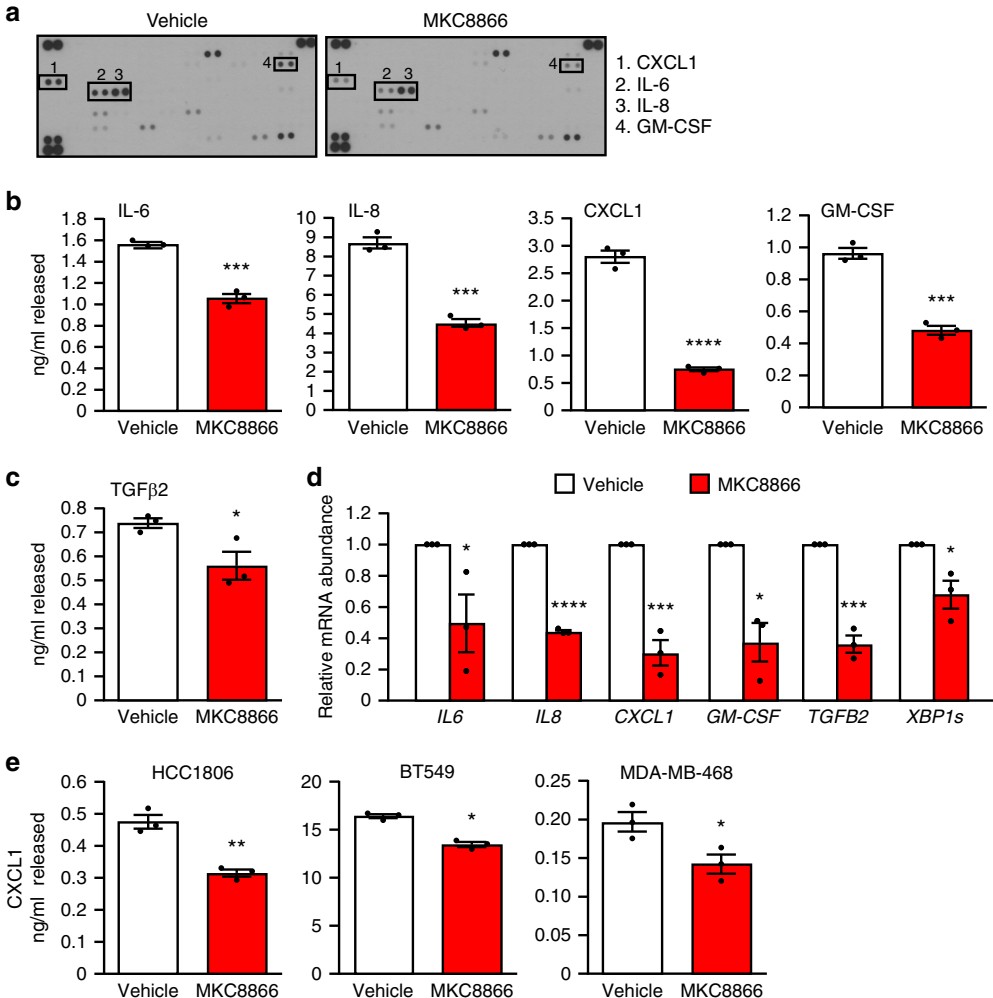

**Fig. 4** IRE1 activity induces production of pro-inflammatory factors. **a–d** MDA-MB-231 cells cultured in medium containing 2% serum were treated with 20 μM MKC8866 or vehicle alone for 48 h after which cells and conditioned medium were collected. **a** Conditioned medium was applied to a Human XL Cytokine Array. Expression profile of cytokines in vehicle alone versus MKC8866-conditioned medium was determined by chemiluminescence. **b**, **c** Cytokine secretion was quantified in conditioned medium using ELISAs selective for IL-6, IL-8, CXCL1, GM-CSF, and TGFβ2 ($n = 3$). **d** mRNA transcript levels of *IL6, IL8, CXCL1, GM-CSF, TGFB2*, and *XBP1s* were quantified by Q-PCR ($n = 3$). **e** CXCL1 quantification in conditioned medium collected from HCC1806, BT549, and MDA-MB-468 cells treated for 48 h in 2% serum-containing medium supplemented with vehicle alone or 20 μM MKC8866 ($n = 3$). Results shown for **a** are representative of two independent experiments. *$P < 0.05$, **$P < 0.01$, ***$P < 0.001$ and ****$P < 0.0001$, based on a Student's $t$ test. Error bars represent s.e.m.

patients. To determine if chemotherapeutics such as paclitaxel impact IRE1 RNase activity, MDA-MB-231 cells were treated with therapeutically relevant concentrations of paclitaxel and IRE1 RNase activity was assessed. Paclitaxel concentrations as low

as 10 nM increased IRE1 RNase activity as demonstrated by an increase in levels of XBP1s protein (Fig. 5a). Moreover, addition of MKC8866 was sufficient to completely block paclitaxel-induced expression of XBP1s (Fig. 5b). Since our results

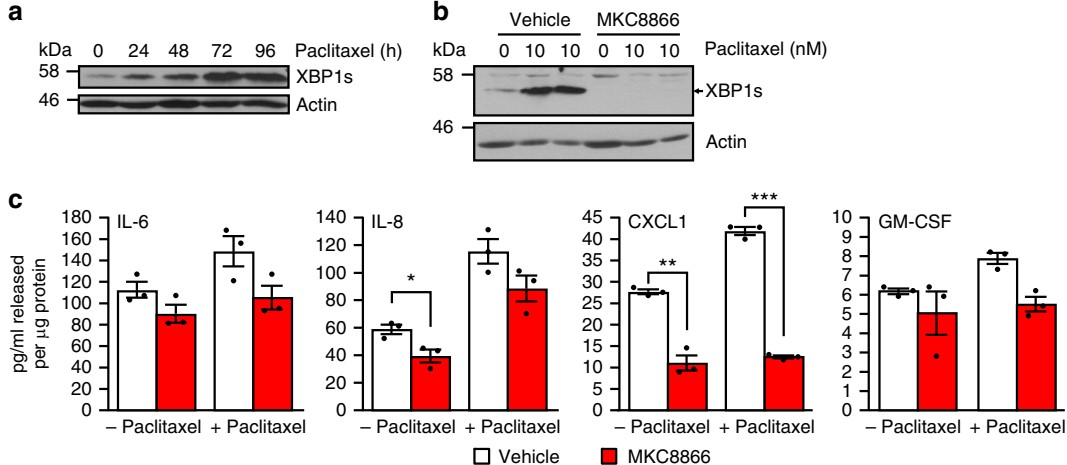

**Fig. 5** Paclitaxel increases IRE1-dependent cytokine secretion. **a** MDA-MB-231 cells were treated with 10 nM paclitaxel for the indicated times, and cell lysates were immunoblotted for XBP1s and Actin. **b** MDA-MB-231 cells were treated with 10 nM paclitaxel in the presence of 20 μM MKC8866 or vehicle (DMSO) for 72 h, cell lysates were harvested and immunoblotted for XBP1s and Actin. **c** MDA-MB-231 cells were treated with 10 nM paclitaxel in combination with DMSO or 20 μM MKC8866 for 72 h in the presence of Boc-D-fmk (40 μM). Following treatment conditioned medium was collected and analyzed by ELISA for secretion of IL-6, IL-8, CXCL1, and GM-CSF. Cells were lysed and protein quantified ($n = 3$). Results shown for **a** and **b** are representative of three independent experiments. $*P < 0.05$, $**P < 0.01$, $***P < 0.001$, based on a Student's $t$ test. Error bars represent s.e.m.

indicated that IRE1 RNase activity exerted control over the secretome and that paclitaxel treatment enhanced IRE1 RNase activity, we investigated whether paclitaxel treatment could increase cytokine production in an IRE1 RNase-dependent manner. We treated MDA-MB-231 cells with vehicle, paclitaxel or a combination of paclitaxel and MKC8866. Following 72 h treatment, cells and conditioned medium were harvested and analyzed for cytokine production. As predicted, we observed a paclitaxel-induced increase in the secretion of IL-6, IL-8, CXCL1, and GM-CSF (Fig. 5c). Co-treatment with MKC8866 reduced paclitaxel-induced increases in CXCL1, GM-CSF, IL-6, and to a lesser extent IL-8 secretion (Fig. 5c).

Studies have shown that chemotherapy, while effective in the short term, can lead to tumor relapse in the longer term, with increases in pro-tumorigenic secreted factors considered key elements in this process[18,19]. To assess the longer term impact of paclitaxel on MDA-MB-231 cells we treated cells with a low dose (10 nM) of paclitaxel for 72 h, after which paclitaxel was removed, cells washed and left to recover in fresh medium containing vehicle alone or MKC8866. After 72 h of recovery, conditioned medium was collected and cytokine levels were assessed. Addition of MKC8866 post-paclitaxel treatment significantly reduced CXCL1 and IL-8 levels compared to vehicle-only controls (Fig. 6a). Since chemotherapy-induced increases in pro-tumorigenic cytokines have been linked to expansion of tumor-initiating cell populations, we assessed the ability of paclitaxel-treated cells incubated with either MKC8866 or vehicle alone during the recovery phase to form mammospheres (a functional readout of tumor-initiating cell expansion). Following 72 h of recovery, cells were counted, equal numbers seeded out onto low-adherence plates, and 5 days later, mammospheres greater than 40 μm were counted and mammosphere-forming efficiency was calculated. As previously reported[18], treatment with paclitaxel significantly increased the mammosphere-forming efficiency of MDA-MB-231 cells (Fig. 6b). Addition of MKC8866 post-paclitaxel treatment substantially reduced mammosphere formation when compared to vehicle-only treated controls, suggesting an important role for IRE1 RNase signaling in this process (Fig. 6b). Since we had observed a reduction in the levels of CXCL1 and IL-8 in conditioned medium post-paclitaxel

treatment (Fig. 6a), we asked if these cytokines contributed to paclitaxel-induced mammosphere formation in MDA-MB-231 cells. To answer this, we depleted CXCL1 and IL-8 levels through addition of neutralizing antibodies during the 72 h recovery phase post-paclitaxel treatment and assessed the ability of cells to form mammospheres. Addition of neutralizing antibodies against either IL-8 or CXCL1 blocked the ability of paclitaxel-treated MDA-MB-231 cells to form mammospheres (Fig. 6c, Supplementary Fig. 4g-i). In addition to neutralizing CXCL1 and IL-8, we carried out a reciprocal experiment where we assessed the ability of exogenous CXCL1 and IL-8 to overcome MKC8866-mediated suppression of mammosphere formation. Combination of MKC8866 with recombinant CXCL1 or IL-8 during the 72 h recovery phase partially reversed MKC8866-mediated inhibition of mammosphere formation (Fig. 6d) further underscoring the importance of these pro-tumorigenic cytokines.

**MKC8866 enhances the effectiveness of paclitaxel in vivo.** To determine the efficacy of MKC8866 treatment in vivo, MDA-MB-231 tumor xenografts were established in athymic nude mice. Once tumors had reached a palpable size (225–250 mm³), animals were randomized into treatment groups and treated with vehicle alone, 300 mg kg⁻¹ MKC8866 alone, 10 mg kg⁻¹ paclitaxel alone or a combination of paclitaxel and MKC8866. Treatments in all groups were administered until tumors reached maximal size (2000 mm³) or on day 60, whichever came first. MKC8866 was well tolerated after 60 consecutive oral doses and, based on pharmacokinetic allometric scaling, systemic exposures were well above anticipated clinical therapeutic levels. Treatment with MKC8866 alone did not attenuate tumor growth compared to vehicle-only controls (Fig. 7a). Analysis of percentage *XBP1* mRNA splicing in those tumors treated with MKC8866 confirmed a reduction in IRE1 RNase activity verifying on-target effect (Fig. 7b). While paclitaxel treatment reduced tumor growth, combination with MKC8866 markedly enhanced the efficacy of paclitaxel. Significantly reduced tumor growth ($P \leq 0.0001$) was observed throughout the 60-day experiment in animals receiving a paclitaxel-MKC8866 combination compared to paclitaxel alone (Fig. 7c). A similar synergistic effect was observed following a

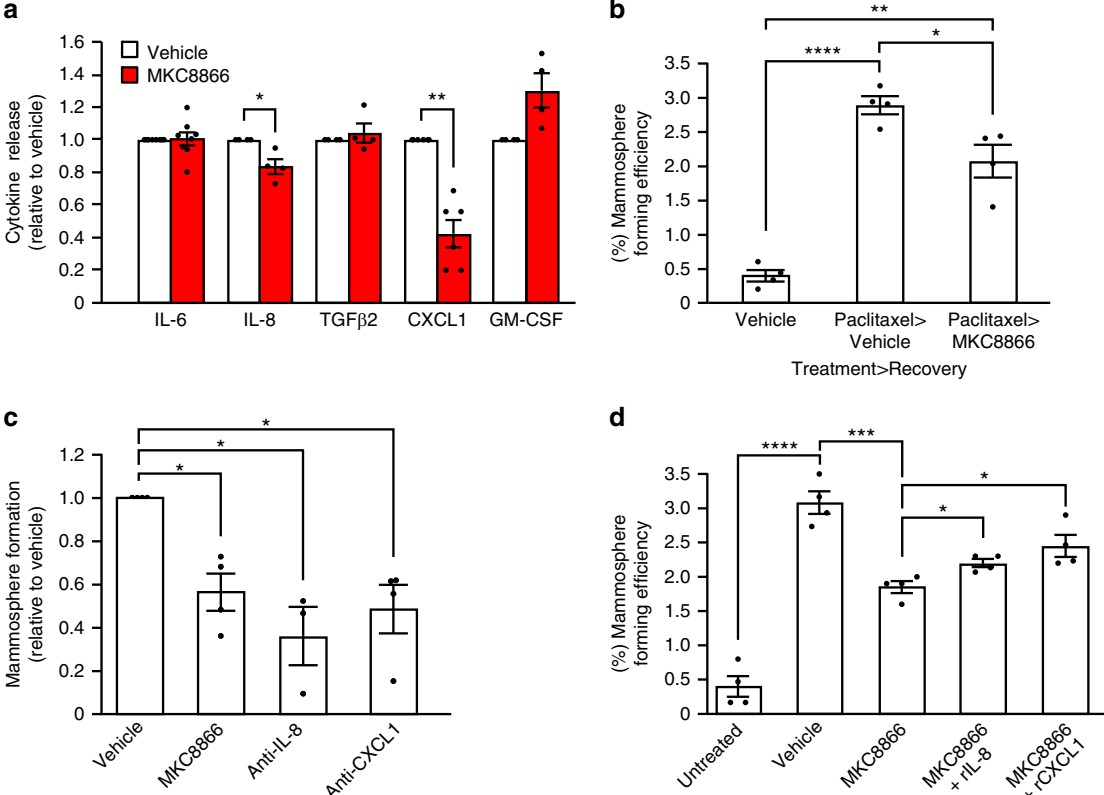

**Fig. 6** Paclitaxel alteration of secretome enhances mammosphere formation. **a–d** MDA-MB-231 cells were treated with paclitaxel (10 nM) for 72 h, after which paclitaxel-containing medium was removed and cells were washed. **a, b** Cells were incubated for a further 72 h in fresh medium containing vehicle alone or MKC8866 (20 μM). **a** Conditioned medium was collected and levels of IL-6 ($n = 7$), IL-8 ($n = 4$), TGFβ2 ($n = 4$), CXCL1 ($n = 6$), and GM-CSF ($n = 4$) were quantified by ELISA. Cytokine release was normalized to that of vehicle-only treated controls. **b** After treatment, cells were seeded at equal densities onto low-adherence plates and mammospheres quantified after a further 5 days ($n = 4$). **c, d** Fresh medium containing vehicle alone, MKC8866 (20 μM), **c** neutralizing antibodies against CXCL1 (10 μg ml$^{-1}$) or IL-8 (500 ng ml$^{-1}$) ($n = 4$), **d** MKC8866 (20 μM) plus recombinant CXCL1 (rCXCL1, 500 pg ml$^{-1}$) or recombinant IL-8 (rIL-8, 3 ng ml$^{-1}$) was added and cells incubated for an additional 72 h. Cells were seeded at equal densities onto low-adherence plates and mammospheres quantified after a further 5 days ($n = 4$). *$P < 0.05$, **$P < 0.01$ and ***$P < 0.001$, based on a Student's $t$ test. Error bars represent s.e.m.

paclitaxel-MKC8866 combination starting on day 14 (or ~700 mm$^3$ tumor volume) ($P \leq 0.001$) or on day 28 (or ~1300 mm$^3$ tumor volume) ($P \leq 0.05$) when compared to paclitaxel alone (Fig. 7c). Examination of *XBP1* splicing in tumors revealed paclitaxel treatment increased IRE1 RNase activity, which was reduced upon combination with MKC8866 (Fig. 7d, Supplementary Fig. 5). The decrease in tumor volume observed following a combination of paclitaxel and MKC8866 also translated to an increase in survival. Mice receiving daily MKC8866 administration in combination with paclitaxel from day 1 to 60, day 14 to 60, and day 28 to 60 displayed significantly longer survival compared to those treated with paclitaxel alone (Fig. 7e).

Since our in vitro studies indicated that IRE1 RNase inhibition by MKC8866 reduced mammosphere formation post-paclitaxel treatment, we tested the outcome of maintaining IRE1 inhibition following paclitaxel withdrawal in vivo. Following MDA-MB-231 tumor formation, mice were treated with paclitaxel alone (7.5 mg kg$^{-1}$) for days 1–10, or a combination of paclitaxel (days 1–10) and MKC8866 (300 mg kg$^{-1}$, days 1–28). After withdrawal of paclitaxel treatment on day 10, an initial reduction in tumor volume was apparent in both treatment groups (Fig. 8). Tumor regrowth, evident after day 18 in those animals receiving no further treatment, was repressed in the treatment group still receiving MKC8866. Tumor regrowth was only apparent in this group following cessation of MKC8866 on day 28 (Fig. 8).

After 28 days of dosing the mice, the maximum systemic concentration of MKC8866 was ~110 μg ml$^{-1}$ as measured by LC/MS/MS with no signs of overt toxicity or significant changes in body weight. Tumor volume measurements revealed 8 out of 10 animals displayed partial tumor regression and 1 animal showed complete tumor regression in the paclitaxel-MKC8866 combination group (Supplementary Table 2). This compared favorably to paclitaxel alone, which had just three partial regressions, one complete regression, and one tumor-free survival observed (Supplementary Table 2). Additional studies are required to fully evaluate tumor growth after treatment is discontinued.

## Discussion

The current dogma regarding IRE1 signaling in cancer is very much aligned with its role as a mediator of the UPR facilitating cell survival under stress conditions. While this is undoubtedly an important function of IRE1 signaling especially early in tumorigenesis[21], numerous other reports have linked IRE1 signaling to facets of tumor biology more aligned with tumor progression, including angiogenesis and metastasis[10,22]. In our system, we identified IRE1 RNase signaling as an important modulator of the secretome in TNBC cells. Through a combination of transcriptomics, Q-PCR, cytokine arrays, and ELISAs we identified IL-6, IL-8, CXCL1, GM-CSF, and TGFβ2

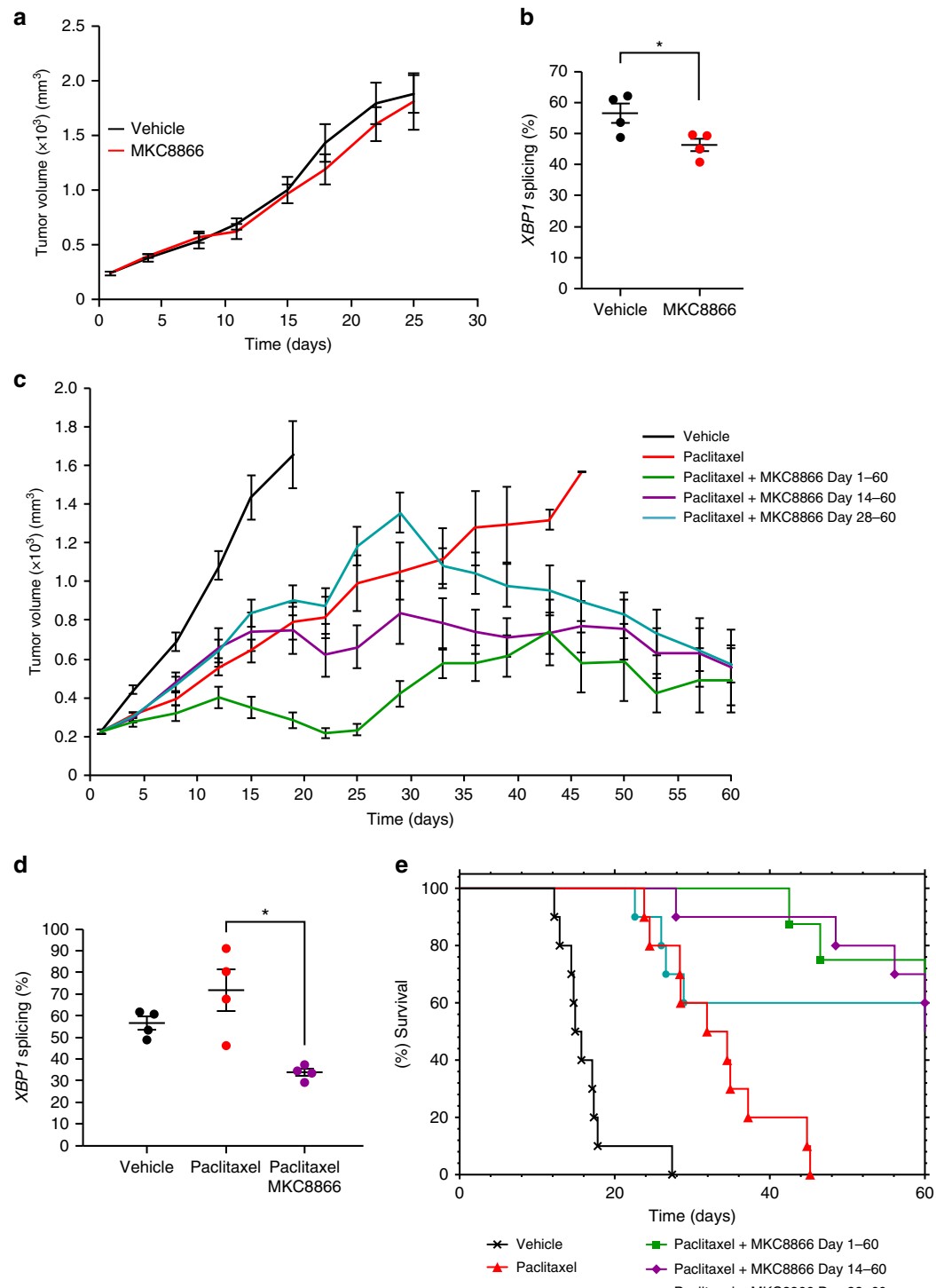

**Fig. 7** MKC8866 enhances the effectiveness of paclitaxel in vivo. Xenografts were established by subcutaneously injecting $5 \times 10^6$ MDA-MB-231 cells into the right flank of female athymic nude mice (Crl:NU(Ncr)-*Foxn1nu*, Charles River). When tumors were palpable (250 mm$^3$) mice were randomized into groups and treatments initiated. **a** Vehicle-only versus MKC8866 (300 mg kg$^{-1}$) daily via oral gavage. Tumor size was assessed every 2–3 days via caliper measurement and tumor volume calculated. By day 25, all tumors had reached their maximum permitted size ($n = 10$ mice per group). **b** Percentage *XBP1* mRNA splicing was determined in vehicle-only versus MKC8866-treated xenografts ($n = 4$ per treatment group). **c** Paclitaxel was administered weekly at 10 mg kg$^{-1}$ by intravenous injection, alone and in combination with MKC8866 administered daily at 300 mg kg$^{-1}$ by oral gavage from day 1 to 60, from day 14 to 60, and from day 28 to 60. Tumor size was assessed every 2–3 days via caliper measurement and tumor volume calculated ($n = 10$ mice per group). **d** Percentage *XBP1* mRNA splicing was determined in vehicle-only, MKC8866-treated and paclitaxel plus MKC8866-treated xenografts ($n = 4$ per treatment group). **e** Kaplan–Meier plot showing survival in animals administered with MKC8866 in combination with paclitaxel (for indicated times) compared to paclitaxel alone or vehicle alone. *$P < 0.05$, based on a Student's *t* test. Error bars represent s.e.m.

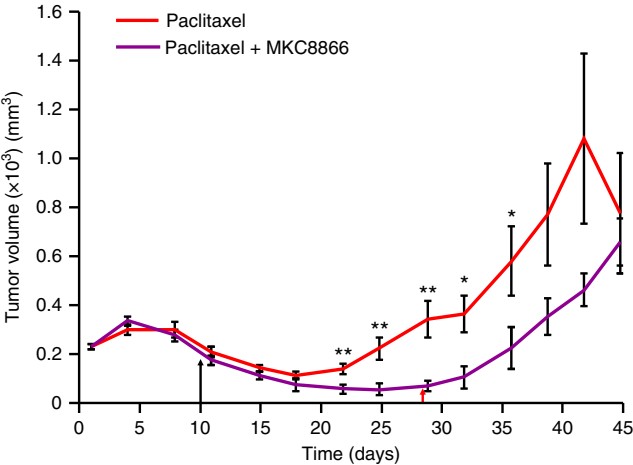

**Fig. 8** MKC8866 reduces tumor regrowth post-paclitaxel withdrawal. Xenografts were established by subcutaneously injecting $5 \times 10^6$ MDA-MB-231 cells into the right flank of female athymic nude mice (Crl:NU(Ncr)-Foxn1nu, Charles River). When tumors were palpable (250 mm$^3$) mice were randomized into groups and treatments initiated. Paclitaxel (7.5 mg kg$^{-1}$ by intravenous injection) was administered every second day until day 10 (last dose indicated by the black arrow) as a single agent or in combination with MKC8866 (300 mg kg$^{-1}$ by oral gavage). MKC8866 treatment was administered daily from day 1 to 28 (last dose indicated by the red arrow). Tumor size was assessed every 2–3 days via caliper measurement and tumor volume calculated ($n = 10$ mice per group). *$P < 0.05$, **$P < 0.01$ based on a Student's $t$ test. Error bars represent s.e.m.

as factors regulated in an IRE1-dependent manner in MDA-MB-231 cells. Previous reports have linked IRE1-dependent signaling, via direct XBP1s transcriptional upregulation, to both IL-6 and IL-8 production in innate immune cells and most recently in melanocytes[23–25] but a role for IRE1 RNase activity in promoting CXCL1, GM-CSF, or TGFβ2 expression has not been previously reported. Cells transfected with XBP1 siRNA displayed decreased transcript levels of all cytokines tested suggesting a direct IRE1–XBP1s signaling mechanism. However, given that XBP1 knockdown also reduced IRE1 transcript levels, further studies are required to determine the exact mechanism. Examination of nuclear-factor-kappa-light-chain-enhancer of activated B cells (NF-κB), signal transducer and activator of transcription 3 (STAT3), and hypoxia-inducible-factor-1α (HIF1α) in MDA-MB-231 cells following MKC8866 treatment did not detect any change in the activation of these transcription factors in MDA-MB-231 cells (Supplementary Fig. 6). Irrespective of the precise signaling pathway our work clearly demonstrates a link between IRE1 RNase activity and the production of soluble factors by TNBC cells.

IL-6, IL-8, CXCL1, GM-CSF, and TGFβ2 are recognized pro-tumorigenic factors associated with cancer progression[18,26–28]. Elevated serum and tissue levels of IL-6 and IL-8 are markers of poor clinical outcome in breast cancer[29–32] and, along with CXCL1, both IL-6 and IL-8 have been implicated in TNBC tumor progression in vivo[26,28]. The tumor secretome has been linked to the recruitment of diverse cell types establishing a pro-tumorigenic microenvironment. CXCL1 and GM-CSF have been reported to recruit myeloid-derived suppressor cells (MDSCs), a cell type connected to tumor angiogenesis and T cell immunosuppression[28,33] while IL-8 and IL-6 have been linked to the recruitment of mesenchymal stem cells and neutrophils amongst others. This suggests that inhibiting IRE1 RNase activity could be an effective way of limiting the impact of the tumor cell

secretome on both the tumor itself and the wider tumor micro-environment. Indeed, recent work in glioblastoma multiform supports this viewpoint. Lhomond and colleagues linked enhanced IRE1–XBP1s signaling to the promotion of angiogenesis, invasion, and macrophage recruitment[34].

While undoubtedly important in cancer progression, the impact of the tumor secretome is particularly pertinent post therapy. Currently, neo-adjuvant chemotherapy is the only treatment option available to TNBC patients. Although successful in the short term, a large number of patients relapse within 1–3 years[35] with cancer stem cell expansion, a key determinant in tumor relapse[36,37]. This subset of tumor cells, which are highly resistant to chemotherapy and radiation, drive tumor re-establishment resulting in drug resistant, rapidly proliferating and highly metastatic tumors refractory to treatment. Therapy-induced changes in the tumor secretome have been identified as a driver of cancer stem cell expansion. In breast cancer specifically, cancer stem cell expansion post therapy has been reported[37,38] and shown to be dependent on the production of several pro-tumorigenic factors including IL-6, IL-8, and TGFβ[18,19,39]. As such, developing therapies to reduce cancer stem cell expansion is key to limiting tumor re-emergence post therapy. Several studies have already illustrated the potential of targeting pro-tumorigenic factors post therapy in breast cancer. Inhibitors of C–X–C motif chemokine receptor 1 (CXCR1), the IL-8 receptor, and antagonistic TGFβ antibodies have been demonstrated to reduce cancer stem cell expansion and limit tumor relapse post therapy[18,39]. Similar to published findings, we observed increased cytokine production post-paclitaxel treatment in MDA-MB-231 cells. Our study takes this further, demonstrating a link between increased pro-tumorigenic cytokine production and a therapy-induced increase in IRE1 RNase activity. Combining MKC8866 with paclitaxel inhibited splicing of XBP1 and reduced production of pro-tumorigenic cytokines in vitro. Mammosphere formation, a widely used functional readout for tumor-initiating cells, was significantly increased in MDA-MB-231 cells following paclitaxel treatment. Addition of MKC8866, post-paclitaxel treatment, reduced mammosphere formation indicating a requirement for IRE1 RNase activity. Analysis of conditioned medium revealed MKC8866 addition reduced the levels of both CXCL1 and IL-8. Moreover, addition of neutralizing antibodies against IL-8 or CXCL1 post paclitaxel also reduced mammosphere formation. These data suggest reduced mammosphere formation in MKC8866-paclitaxel-treated cells is a likely consequence of decreased cytokine production. Similar to our findings, Chen and colleagues, using an inducible shRNA-mediated knockdown of *XBP1*, recently linked XBP1 expression to cancer stem cell expansion and tumor relapse post therapy, although a role for the secretome in this process was not extensively explored[10]. While genetic knockdown is a powerful experimental tool, it is not a viable therapeutic strategy easily translatable to patients unlike a small molecule inhibitor. In vivo, we found daily MKC8866 administration to be well tolerated in mice with no toxicity evident after 60 consecutive daily oral doses at 300 mg kg$^{-1}$. Although not effective as a single agent in the TNBC xenograft model, MKC8866 significantly enhanced paclitaxel-mediated repression of tumor growth. Additionally, maintenance of IRE1 RNase inhibition post-paclitaxel withdrawal sustained suppression of tumor regrowth, confirming in vivo, that blocking IRE1 RNase may increase the efficacy of chemotherapeutic agents such as paclitaxel.

Our study supports IRE1 as a therapeutic target in TNBC and illustrates the therapeutic potential of small molecule IRE1 RNase inhibitors in TNBC treatment. Recently published findings by Zhao and colleagues further support this view, demonstrating pharmacological inhibition of the IRE1–XBP1 pathway

suppresses MYC-driven breast cancers[40]. To extrapolate our in vitro findings to primary patient-derived samples, we generated a putative IRE1 gene signature reflective of IRE1 RNase activity. When applied to TCGA breast cancer gene expression data sets, this gene signature identified two distinct cohorts reflective of IRE1 RNase high and IRE1 RNase low activities. Examination of these groupings revealed that breast cancers characterized by a high IRE1 gene signature associated with basal-like breast cancers and exhibited increased expression of pro-inflammatory factors when compared to those with low IRE1 gene signature. These data, in conjunction with our in vitro cell line findings, strongly support the hypothesis that an elevated IRE1 RNase activity is associated with basal-like/TNBC. The development of genetic signatures such as this could enable the identification of breast cancer patients most likely to benefit from treatment with IRE1 RNase inhibitors and act as a companion diagnostic. In conclusion, our work demonstrates a role for IRE1 signaling as an important regulator of the TNBC cell secretome and provides compelling evidence to support the use of IRE1 RNase inhibitors in combination with chemotherapeutics for the treatment of TNBC.

## Methods

**Cell culture and treatments**. MCF10A (ATCC) cells were maintained in DMEM/F-12 (Gibco, 11320-074) supplemented with 5% horse serum (Sigma-Aldrich, H1270), 20 ng ml⁻¹ epidermal growth factor (PeproTech, AF-100-15), 0.5 µg ml⁻¹ Hydrocortisone (Sigma-Aldrich, H0888), 100 ng ml⁻¹ Cholera toxin (Sigma-Aldrich, C8052), 10 µg ml⁻¹ insulin (Sigma-Aldrich, I1882), 50 U ml⁻¹ penicillin, and 50 µg ml⁻¹ streptomycin (Sigma-Aldrich, P0781). MCF7 cells (ECACC) were cultured in DMEM high glucose (Sigma-Aldrich, D6429), SKBR3 (ECACC) cells in McCoys5A (Sigma-Aldrich, M9309), HCC1806 cells (ATCC), and BT549 (ATCC) in RPMI-1640 medium (Sigma-Aldrich, R0883) supplemented with 10% fetal bovine serum (FBS) (Sigma-Aldrich, F7524), 50 U ml⁻¹ penicillin, 50 µg ml⁻¹ streptomycin (Sigma-Aldrich, P0781), and 2 mM L-glutamine (Sigma-Aldrich, G7513). MDA-MB-231 cells obtained from ATCC and ECACC were used in this study. MDA-MB-231 cells were cultured in DMEM high glucose (Sigma-Aldrich, D6429) supplemented with 10% FBS, 50 U ml⁻¹ penicillin, 50 µg ml⁻¹ streptomycin, and 2 mM L-glutamine. The data from both cell clones were similar. Cells from ATCC were tested and were mycoplasma negative. HEK293T cells were from ATCC and were cultured in DMEM high glucose supplemented with 10% FBS, 2 mM L-glutamine, 50 U ml⁻¹ penicillin, and 50 µg ml⁻¹ streptomycin. All cells were cultured at 37 °C, 5% CO₂ in a humidified incubator and seeded at an appropriate number 24 h prior to treatment. Cells were treated with the indicated concentrations of MKC8866 (Fosun Orinove PharmaTech Inc.), Tunicamycin (Tm) (Sigma-Aldrich, T7765), Thapsigargin (Tg) (Sigma-Aldrich, T9033), Paclitaxel (Sigma-Aldrich, T7402), Boc-D-fmk (Biovision, 1160-5), Etoposide (Sigma-Aldrich, E1383), or an equal volume of DMSO (Sigma-Aldrich, D2650). Neutralizing antibodies against IL-6 (R&D Systems, MAB206), IL-8 (R&D Systems, MAB208), CXCL1 (R&D Systems, MAB275), and GM-CSF (R&D Systems, MAB215) were used at 0.5 µg ml⁻¹, while TGFβ2 neutralizing antibody (R&D Systems, AF302) was used at 1.12 µg ml⁻¹.

**Western blotting**. Cells were washed once in ice-cold phosphate-buffered saline (PBS) and lysed in whole cell lysis buffer (2% sodium dodecyl sulfate (SDS), 50 mM Tris HCl pH 6.8, 5% glycerol, 0.05% bromphenol blue, 357 mM β-mercaptoethanol) or radioimmunoprecipitation assay (RIPA) buffer (0.1% SDS, 1% NP-40, 0.5% sodium deoxycholate, 50 mM Tris-HCl pH 8.8, 150 mM NaCl) after indicated treatments and cell lysate boiled at 95 °C for 5 min. Protein samples were separated on an SDS polyacrylamide gel, transferred onto nitrocellulose membrane (Amersham Protran 0.2 10600001) and blocked with 5% milk in PBS-0.1% Tween. For detection of protein expression the following antibodies were used: Actin (Sigma-Aldrich, A-5060, 1:5000), XBP1 (Abcam, 37152, 1:1000), XBP1s (Biolegend, 647502, 1:1000), PERK (CST, 3192, 1:1000), CHOP (CST, 2895, 1:1000), ATF6 (CosmoBio, AM-73-500-B, 1:1000), phospho-p65 (CST 3033, 1:1000), total-p65 (CST 8242, 1:5000), phospho-STAT3 (SantaCruz, 8059, 1:1000), total-STAT3 (Santa Cruz, 482, 1:1000), β-catenin (CST 8480, 1:1000), FOXO1 (CST 2880, 1:1000), and HIF1α (Novus Biologicals, NB100-479, 1:1000). Anti-rabbit (111-035-003) and anti-mouse (115-035-003) HRP-conjugated secondary antibodies were purchased from Jackson Immunoresearch and the signal was visualized using western blotting luminol reagent (SantaCruz, sc-2048). Uncropped western blot images are shown in Supplementary Fig. 7.

**EdU incorporation assay**. S phase cells were determined using EdU (Berry & Associates PY7563) according to manufacturer's instructions. Post-treatment cells were trypsinised, washed with PBS, and fixed in 70% ice-cold ethanol for 1 h before

storing at −20 °C. After thawing cells were washed in PBS, resuspended in 1 ml of Click cocktail (10 mM sodium ascorbate, 100 µM 5′ fluorescein isothiocyanate azide and 2 mM copper II sulfate) and incubated in the dark at room temperature for 30 min. The signal was quenched by adding 10 ml PBS, 0.5% Tween-20, and 1% bovine serum albumin (BSA) for 10 min at room temperature. Samples were then washed with PBS and analyzed on the FL1 channel of a FACSCalibur flow cytometer (Becton Dickinson).

**Propidium iodide assessment of cell death**. Membrane permeability was assessed using propidium iodide (PI) staining. Briefly, cells were harvested by trypsinization and incubated 15 min at 37 °C to restore membrane integrity. Cells were collected by centrifugation, resuspended in PBS, stained with 0.6 µg ml⁻¹ of PI (Sigma-Aldrich 81845), and analyzed using a FACSCalibur flow cytometer (Becton Dickinson).

**Generation of conditioned medium**. MDA-MB-231 cells were cultured in 2% serum-containing medium in the presence of 20 µM MKC8866 or an equal volume of vehicle (DMSO). After 48 h, the supernatant was removed, filtered, and the resultant conditioned medium was used for further experiments. In the case of neutralizing antibody experiments, a portion of the conditioned medium post neutralization was analyzed by ELISA to confirm successful neutralization.

**Human XL Cytokine Array**. We used a Human XL Cytokine Array kit (R&D systems, ARY022) as per manufacturers' instructions. In brief, membranes were blocked with Buffer 6 for 1 h at room temperature and incubated with conditioned medium overnight at 4 °C with gentle agitation. The following day, membranes were washed three times for 10 min with wash buffer. Detection antibody cocktail was added to each membrane for 1 h at room temperature followed by washing. Streptavidin-HRP (2 ml) was added to each membrane and incubated for 30 min at room temperature. Membrane washes were repeated and signals were visualized by addition of Chemi Reagent Mix (2 ml).

**ELISA**. IL-6 (DY206), IL-8 (DY208), CXCL1 (DY275), GM-CSF (DY215), and TGFβ2 (DY302) DUOSET ELISA's were purchased from R&D Systems and carried out as per manufacturer's instructions.

**RNA extraction, PCR, and Q-PCR**. Total RNA was isolated using TRI Reagent (Sigma-Aldrich, T9424) according to the manufacturer's instructions. In total, 500–2000 ng of RNA was reverse transcribed using Superscript II (Invitrogen 18064014). For standard PCR, products were visualized using 1% agarose gels. Q-PCR reactions were performed using Takyon ROX Master Mix (Eurogentec UFRP5XC0501) and the StepOne Plus platform (Applied Biosystems). Target transcript levels were normalized to *GAPDH*, and relative abundance was determined using the ΔΔCt method. Transcript-specific TaqMan assays were purchased from Integrated DNA Technologies. For tumor xenografts, *XBP1u* and *XBP1s* transcript levels were normalized to the average Ct of control gene, *ACTB* using the ΔΔCt method. MDA-MB-231 cDNA was used as the control sample for ΔΔCt calculations. Percentage *XBP1* mRNA splicing was determined by calculated as *XBP1s*/(*XBP1s* + *XBP1u*) × 100. Sequences of primers and probes used are detailed in Supplementary Table 3.

**Q-PCR to determine relative *XBP1* splicing in patient samples**. The Galway University Hospitals Clinical Research Ethics Committee approved the use of human tissue samples following informed patient consent. Patients provided written informed consent for use of samples, and work was performed according to the principles of the Declaration of Helsinki. Breast tissue samples (basal (*n* = 5), luminal (*n* = 4), TAN (*n* = 4)) were harvested in theater at University Hospital Galway. Samples were preserved by immediate immersion in RNAlater® (Sigma-Aldrich R0901) and subsequently stored at −80 °C until required for RNA extraction. RNA quality was determined by resolving at least 250 ng of total RNA on a 1% sodium borate agarose gel, and samples displaying degradation were excluded from the study. An aliquot of 500 ng of total RNA was reverse transcribed as described above. Total *XBP1* and *XBP1s* transcript levels were normalized to the average Ct of control genes, *PPIA* and *MRPL19* using the ΔΔCt method. A pool of cDNA was used as an inter-plate/run control and as the control sample for ΔΔCt calculations. Results are displayed as: relative *XBP1s* abundance/relative *XBP1* total abundance.

**Transient knockdown and overexpression**. For knockdown, MDA-MB-231 cells were transfected with 25 nM of Dharmacon On-Target SMARTpool Plus siRNA targeting XBP1 (L-009552-00), IRE1 (L-004951-02), or non-targeting control (NC) siRNA (D-001810-01-20) using Dharmafect 4 (Dharmacon T-2004-02) according to the manufacturer's instructions.

**Stable shRNA knockdown**. pLKO control vector (SHC001) and human *XBP1* (SHCLND-NM_005080) lentiviral shRNA constructs were purchased from Sigma-Aldrich. Lentivirus was generated by co-transfecting the above plasmids with

second-generation lentivirus-packaging system (Addgene, pMD2.G cat. no. 12259, psPAX2 cat. no. 12260, pRSV-Rev cat. no. 12253) using JET PEI transfection reagent (Polyplus Transfection, cat. no. 101-01N) into HEK 293T cells. Virus-containing supernatant was harvested and filtered through 0.22 μm filter (Sarstedt Filtropur 83.1826.001). Cells were transduced with this media in presence of 5 μg ml$^{-1}$ polybrene (Merck Millipore TR-1003-G). XBP1 shRNA cells were selected for 72 h in 2 μg ml$^{-1}$ of puromycin (Sigma-Aldrich, P8833).

**Microarray analysis**. MDA-MB 231 cells were seeded at $5 \times 10^5$ cells per T25 flask. Cells were treated with vehicle (DMSO) or MKC8866 (20 μM) for 4 and 24 h. RNA was extracted using a combination of phenol-chloroform extraction (TRI reagent T9424), and Qiagen RNeasy column (74104) extraction methods. A portion of the RNA was taken for pre-analysis. Inhibition of *XBP1* splicing was confirmed by RT-PCR before the samples were sent to EMBL for analysis. RNA quality was determined by amplifying 5′ and 3′ ends of the *GAPDH* transcript upon OligoDt-primed reverse transcription, and by capillary electrophoresis upon receipt at EMBL. The experiment was performed in triplicate. MicroArray analysis was performed on Affymetrix GeneChip Human Transcriptome Array 2.0 in at the EMBL genetics core facility in Heidlberg, Germany. The generated data have been deposited in the Gene Expression Omnibus (GEO) and are publicly available under the GEO accession number GSE99766. The data were analyzed using GeneSpring GX software (Life Sciences Informatics, Agilent) whereby genes with *P* value below 0.05 and absolute fold change greater than 1.3 were considered differentially expressed between the two conditions and selected as potential IRE1 markers for further analysis. Initial candidate markers were subsequently curated in a training data set by utilizing them to hierarchically cluster, in a semi-supervised manner, a panel of 27 breast cancer line gene expression data sets (GEO accession GSE50832)[41]. Hierarchical clustering utilizing the 401 candidate markers revealed two robust groups of genes that either negatively or positively correlated with expected IRE1 activity. Genes strongly associated with either group were further prioritized using BioInfoMiner tool to generate an 83-gene signature predictive of IRE1 activity (Supplementary Table. 1). This IRE1 83-gene signature was then utilized for unsupervised clustering of a cohort of 595 breast cancer tumors from the publicly available The Cancer Genome Atlas (TCGA) database[42]. Breast cancer tumor groups with gene expression profiles corresponding with high ($n = 63$) and low ($n = 79$) IRE1 activity (as predicted by the IRE1 gene signature) from both ends of the spectrum were taken for further analysis and comparison. *P* values for box plots were calculated using a two-tailed *t* test with Welch's correction and performed with GraphPad Prism software (GraphPad Software, San Diego, CA, USA). GSEA between the two groups was performed with GSEA software (Broad Institute), using transcriptome profiling data from the high IRE1 and low IRE1 tumor sets (TCGA) and pre-defined gene signatures from the Molecular Signature Database (MSigDB). The Pearson correlation coefficient was calculated between predicted IRE1 activity score and every other gene, across the cohort of 595 patient samples from TCGA, resulting in a list of genes ranked according to their correlation with IRE1 activity. GSEA with Gene Ontology terms (GO) of the ranked list was performed with GSEA software (Broad Institute). GS enrichment scores were normalized using 1000 gene permutations and GO terms with false discovery rates under 0.25 were considered significantly enriched.

**Mammosphere formation assay**. MDA-MB-231 cells were treated with 10 nM paclitaxel for 72 h. Cells were washed once with complete growth medium then allowed to recover in complete medium containing vehicle (DMSO), MKC8866 (20 μM), IL-8 (500 ng ml$^{-1}$) neutralizing antibody, CXCL1 (10 μg ml$^{-1}$) neutralizing antibody, recombinant CXCL1 (500 pg ml$^{-1}$), or recombinant IL-8 (3 ng ml$^{-1}$), as indicated for a further 72 h. After recovery cell viability was determined using trypan blue staining. Viable cells from each treatment were seeded in triplicate at $1 \times 10^3$ cells per well in 96 well ultra-low attachment surface plates (Corning, 10554961) in DMEM/F12 medium supplemented with B27-supplement (ThermoFisher, 12587010) and 20 ng ml$^{-1}$ epidermal growth factor. Completely untreated cells were also seeded as a control. Mammospheres measuring >40 μm were quantified in five fields per well, and mammosphere formation efficiency (%) was determined using the following formula: (number of mammospheres > 40 μm/ number of cells seeded) × 100.

**Immunohistochemistry**. The study was conducted with human samples and clinical data of Rennes Biobank Breast Cancer Collection (BRIF number: BB-0033-00056) certified NF S96900 for receipt preparation preservation and provision of biological resources. All patients provided written informed consent to the use of surgical specimens and clinic-pathological data for research purposes (as required by the French Committee for the Protection of Human Subjects (CCPPRB)). Rennes CCPPRB approved the use of tumor tissues for this study (6 May 2013: no. 357/2013). Collection of tumors was approved by French Minister of higher education and research (no. AC-2008-141). Samples were embedded in paraffin and cut into 3 μm sections. Ventana OMNIMap (ROCHE) detection kit procedure was optimized on the Discovery instrument and was preset. Applications of the Ventana High Temperature Liquid Coverslip (LCS, cat. no. 650010, Ventana) occurred throughout the automated protocol as appropriate.

Likewise, the slides were rinsed between steps with Ventana Tris-based Reaction buffer. Following deparaffination with Ventana EZ Prep at 75 °C for 8 min, antigen retrieval was performed using Ventana proprietary, Tris-based buffer solution CC1 (pH 8). Endogenous peroxidase was blocked with Inhibitor-D 3% $H_2O_2$ for 10 min at room temperature. After rinsing, slides were incubated at 37 °C for 60 min with a 1:1000 dilution of mouse polyclonal XBP1[43], rabbit polyclonal IL-8 (Santa Cruz SC-7922, 1:50), or rabbit polyclonal CXCL1 (R&D systems MAB4531, 1:50). Tissue sections were incubated for 30 min with conjugated secondary antibody and visualized following substrate addition. Immunohistochemistry staining was quantified by an independent investigator using a blinded approach according to the methodology outlined previously[44].

**In vivo MDA-MB-231 xenograft model**. All animal experiments were performed in accordance with the ethical guidelines of the IACUC committee at Charles River Laboratories, Piedmont, South Carolina (Approved study protocol IACUC ASP #: 980701). Female athymic nude mice (Crl:NU(Ncr)-*Foxn1nu*, Charles River) were implanted subcutaneously in the right flank with $5 \times 10^6$ MDA-MB-231 cells (0.1 ml cell suspension in PBS). Mice with tumors measuring between 225 and 250 mm$^3$ were randomized into six treatment groups consisting of ten mice with individual tumor volumes ranging from 196 to 288 mm$^3$ and group mean tumor volumes from 225 to 227 mm$^3$ (considered day 1 of treatment). Change in tumor volume was monitored by calipers two times per week with tumor volume calculated as $V = (L \times S2)/2$ by measuring the long (L) and short (S) axes of tumors. Paclitaxel (Lot CP2N10007) was purchased as a dry powder from Phyton Biotech, LLC (Fort Worth, TX). A 10 mg ml$^{-1}$ paclitaxel stock solution in 50% ethanol: 50% Cremophor EL was prepared and stored at room temperature protected from light prior to dosing. On each day of dosing, an aliquot of the paclitaxel stock was diluted with 5% dextrose in water (D5W) to yield a 1.0 mg ml$^{-1}$ paclitaxel dosing solution in a vehicle consisting of 5% ethanol: 5% Cremophor EL: 90% D5W (Vehicle 1) which provided the 10 mg kg$^{-1}$ dose in a 10 ml kg$^{-1}$ dosing volume. Mice were administered 10 mg kg$^{-1}$ paclitaxel weekly by intravenous injection. The IRE1 inhibitor, MKC8866, was administered at a dose volume of 10 ml kg$^{-1}$ from a 30 mg ml$^{-1}$ suspension in 1% microcrystalline cellulose in a simple sugar at 300 mg kg$^{-1}$ daily by oral gavage (Vehicle 2). Treatment groups were as follows: For Group 1, the paclitaxel vehicle was administered intravenously weekly and the MKC8866 vehicle was administered orally daily throughout the course of the study. For Groups 2–6, paclitaxel was administered weekly throughout the course of the study. In combination with paclitaxel, MKC8866 was also administered orally daily from day 1 to 28 (Group 3), from day 14 to 60 (Group 4), from day 28 to 60 (Group 5), and from day 1 to 60 (Group 6). Treatments in all groups were administered until tumors reached maximal size or day 60, whichever came first.

For the xenograft regrowth post-paclitaxel in vivo experiments, female athymic nude mice (Crl:NU(Ncr)-*Foxn1nu*, Charles River) were implanted subcutaneously in the right flank as described above. Following establishment of palpable tumors, mice were randomized into treatment groups consisting of 10 mice per group with group mean tumor volumes from 227 to 230 mm$^3$ (considered day 1 of treatment). A 7.5 mg ml$^{-1}$ paclitaxel stock solution in 50% ethanol: 50% Cremophor EL was prepared and stored at room temperature protected from light prior to dosing. On each day of dosing, an aliquot of the paclitaxel stock was diluted with 5% dextrose in water (D5W) to yield a 0.75 mg ml$^{-1}$ paclitaxel dosing solution in a vehicle consisting of 5% ethanol: 5% Cremophor EL: 90% D5W (Vehicle 1) which provided the 7.5 mg kg$^{-1}$ dose in a 7.5 ml kg$^{-1}$ dosing volume. Mice were administered 7.5 mg kg$^{-1}$ paclitaxel once every other day for five doses by intravenous injection. MKC8866 was administered daily for 28 days at a dose volume of 10 ml kg$^{-1}$ from a 30 mg ml$^{-1}$ suspension in 1% microcrystalline cellulose in a simple sugar at 300 mg kg$^{-1}$ daily by oral gavage (Vehicle 2). Group 1 received paclitaxel (7.5 mg kg$^{-1}$) alone while Group 2 received paclitaxel (7.5 mg kg$^{-1}$) plus 300 mg kg$^{-1}$ MKC8866.

If, during the course of the study, tumors became necrotic or if measurement of the tumor in two dimensions was not possible using calipers, measurement was stopped. Mice were observed frequently for health and overt signs of any adverse treatment-related side effects, and noteworthy clinical observations were recorded. Individual body weight loss was monitored per protocol, and any animal whose weight exceeded the limits for acceptable body weight loss was euthanized. Acceptable toxicity was defined as a group mean body weight loss of <20% during the study and not more than one treatment related death among ten treated animals, or 10%. Any dosing regimen resulting in greater toxicity was considered above the maximum tolerated dose.

**Statistical analysis**. Assumptions concerning the data (normal distribution and similar variation between experimental groups) were examined for appropriateness before statistical tests were conducted. Statistical analysis was carried out using pairwise single factor ANOVA, unpaired two-tailed Student's *t* test, or two-tailed *t* test with Welch's correction as indicated. Values with $P < 0.05$ are considered statistically significant.

**Data availability**. Microarray data supporting the findings of this study have been deposited in the Gene Expression Omnibus and are publicly available under the GEO accession number GSE99766.

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

## Acknowledgements

The authors wish to acknowledge the expertise provided by Charles River Study Director, Kay Mesham, and the NCBES Flow Cytometry, Genomic and Screening core facilities at the National University of Ireland Galway which are funded by NUI Galway and the Irish Government's Programme for Research in Third Level Institutions, Cycles 4 & 5, National Development Plan 2007-2013. We thank Dr. Sandra Healy for critical review of the manuscript and Gwénaële Jégou for her assistance with analysis of in vivo samples. We are grateful to H2P2 facility of Biosit (Rennes, France) for its technical assistance with immunohistochemistry studies. Microarray analysis was carried out in collaboration with Genomics Core Facility EMBL. The work in our groups are funded by Breast Cancer Campaign grant (2010NovPR13, 2015MaySP550), Health Research Board (grant number HRA-POR-2014-643), Belgium Grant (IAP 7/32), Science Foundation Ireland (SFI) grant co-funded under the European Regional Development Fund (grant number 13/RC/2073), SFI Starting Investigator Grant (grant number 15/SIRG/3528), EU H2020 MSCA ITN-675448 (TRAINERS), EU H2020 MSCA RISE-734749 (INSPIRED), and Institut National du Cancer (INCa; PLBIO: 2017-148). P.C. was funded by an Irish Cancer Society Scholarship (CRS11CLE) and Thomas Crawford Hayes Funds of NUIG. E.P.M. was funded by the Irish Research Council Employment Based Programme Scholarship Scheme (BPPG/2014/57). K.M. was funded by an Irish Research Council Fellowship (grant number GOIPD/2014/53).

## Author contributions

S.E.L., E.P.M. and P.C. designed, performed and analyzed experiments. S.G. conducted drug formulation and in vivo studies, determined compound pharmacokinetics, designed and oversaw in vivo experiments and helped with data analysis described in Figs. 7 and 8. K.M. helped with compilation of data and manuscript preparation. A.A. performed data analysis described in Fig. 3 and Supplementary Fig. 3. E.C. oversaw a portion of the work and provided critical discussion of the work. R.M.D. oversaw collection and preparation

of samples used in Fig. 1c. A.O. helped with analysis of data associated with Fig. 3b. P.L. and F.G. provided access to patient material used in Supplementary Fig. 5. E.C.M., B.L., and J.O. designed and performed experiments. J.B.P. and Q.Z. provided access to MKC8866 and critical feedback on the study. R.J. contributed to initiation of the study and helped secure funding. A.M.G. provided critical feedback on the study and contributed to preparation of manuscript and revisions. S.E.L. wrote the manuscript and contributed to study direction. A.S. conceived the study, provided critical feedback, contributed to manuscript preparation and oversaw the research program. All authors listed reviewed the manuscript and provided feedback.

## Additional information

**Competing interests:** A.S., A.M.G., and E.C. are co-founders and shareholders of Cell Stress Discoveries Ltd. S.G., Q.Z., and J.B.P. are employees and shareholders of Fosun Orinove PharmaTech Inc. The remaining authors declare no competing interests.

