## [Peer Review File · Nature Communications]

Reviewers' comments:

Reviewer #1 (Remarks to the Author):

In this manuscript, Logue et al. examine the role of IRE1 α functionally in a number of triple negative breast cancer (TNBC) cell lines and using biomarkers in human tumor samples. As has been previously reported by others (Chen X et al. *Nature* 2014;508:103-7), they find evidence that the IRE1/XBP1 pathway is activated in triple negative breast cancer cell lines and human samples. Moreover, they show that knocking down XBP1 or using a small molecule inhibitor of IRE1 (MKC8866) has anti-proliferative effects on TNBC cell lines grown in culture and is associated with a reduction in the production of a number of pro-inflammatory cytokines (such as CXCL-1 and IL-8). Using xenograft studies in mice, they find that monotherapy with MKC8866 has no significant effects on tumor growth. However, they show that treating cultured TNBC cell lines with paclitaxel upregulates XBP1 and some of the before mentioned cytokines. They then see that the combination of MKC8866 plus paclitaxel results in modestly better anti-tumor growth effects and survival benefits in the xenograft model than paclitaxel alone.

Overall, the findings are interesting and may provide further evidence that IRE1 is a potential target in TNBC, a malignancy for which much better therapies are needed. However, in its present form, the manuscript has a number of major flaws and missing controls that must be addressed before any conclusions can be drawn about the role of IRE1 in this model of TNBC.

Major Concerns:

1) Many of the studies here rely heavily on the use of MKC8866, an aldehyde-based inhibitor that forms a Schiff base with a lysine (K907) in the RNase active site of IRE1 α . In the abstract, the authors inaccurately describe MKC8866 as "novel small molecule" when it was originally reported in 2011 (Volkman K et al. *J Biol Chem* 286; 12743-12755). Hence, it is hardly novel and that term needs to be removed from the manuscript. Second, the authors state in the abstract that this molecule is a "highly selective IRE1 RNase inhibitor." While it is clear that this compound inhibits IRE1 α 's RNase activity in cell culture when used at high concentrations (25 μ M), what is the evidence for selectivity of this compound (showing that PERK and ATF6 are not block is hardly evidence of monoselectivity)? How many other lysines on others proteins do they bind? We have no idea and therefore need to be very cautious with interpreting any results with these compounds (especially as they result to cytotoxicity).

2) Given the questionable selectivity of MKC8866, I very much appreciate the authors' efforts to use shRNA knockdown as a second genetic approach. However, a number of additional controls need to be included. In addition to knocking down XBP1 in MDA-MB-231 cells in Figure 2, they also need to independently knockdown IRE1. Moreover, they then need to test the effect of MKC8866 on the cells knocked down for IRE1 and XBP1. Moreover, they need to also test the effects of MKC8866 on the MDA-MB-231 cells where they over-express XBP1s (these cells should be highly resistant to any effects of this compound).

3) In Figure 2K-L, they need to perform an immunoblot for XBP1s not only after 24 hrs but also in these cells for the duration of the experiment (8 days). We and others have found that many cell types do not allow XBP1s to continue to be over-expressed as a transgene after 24-48hrs through mechanisms that are unclear.

4) In Figure 3, they need to test the effects of over-expressing and blocking the relevant cytokines (CXCL1) on MDA-MB-231 proliferation as assayed in Figure 2.

5) In Figure 4, the IHC images for XBP1s are of such low magnification and quality, that it is impossible to determine if the staining is nuclear or non-specific. They need to provide much better and higher quality images or remove these studies from the manuscript.

6) In Figure 6, they need to test the effects of the neutralizing antibodies on their own (in the absence of paclitaxel).

7) The lack of effect of MKC8866 monotherapy in the xenograft studies is disappointing and challenges the central hypothesis here. They need to harvest the tumors from these MKC8866 treated animals and confirm that IRE1 is indeed inhibited in vivo (XBP1 splicing). Moreover, given the concerns raised above regarding the selectivity of MKC8866, they should also genetically knockdown IRE1 and XBP1 in this cell line prior to injecting it as a xenograft as a second approach.

8) The results in Figure 8 are not at all convincing. Is there a significant difference between the MKC8866 + paclitaxel treated animals vs the paclitaxel alone animals (what is the p-value)? If not, this figure should be removed.

Reviewer #2 (Remarks to the Author):

The manuscript by Logue and colleagues presents some novel and clinically relevant results linking inhibition of the RNase activity of the UPR mediator IRE1 to proliferation arrest and in vivo growth of triple-negative breast cancer (TNBC) cell lines treated with paclitaxel. The link between TNBC and IRE1/XBP-1 activation has been firmly established (Chen et al, Nature, 2014). What is novel in this manuscript is the demonstration that inhibition of IRE1's RNase activity significantly attenuates autocrine and paracrine signaling of pro-tumorigenic cytokines and synergizes with paclitaxel to confer potent anti-tumor effects. Moreover, the finding that TNBC with a higher IRE1 RNase signature, especially with elevated IL-8 and CXCL1 levels are associated with more advanced clinical grade is novel and clinically important. The manuscript is well-written (although some reorganization of sections is recommended) and the data is clear. There are some concerns however regarding the specific mechanism behind the observed anti-cytokine effects and the interpretation of the anti-tumor effects of the MKC8866 inhibitor in combination with paclitaxel.

Major comments:

1. Since a previous publication has linked XBP1 activation in TNBC to regulation of HIF1, it is possible that in vivo some or all the observed growth-inhibitory effects are due to defective angiogenesis. This is also likely because some of the cytokines observed to be affected by the MK inhibitor are potential HIF1 targets (Chen et al, 2014). Analysis of angiogenesis in tumors (e.g., by analyzing microvessel density in tumors), as well as HIF1 levels needs to be analyzed.
2. The in vitro results clearly establish that the MKC8866 inhibitor attenuates TNBC proliferation alone. Yet, in vivo, the MKC8866 drug has no effect, but does synergize with paclitaxel to inhibit tumor growth. Why the discrepancy? Has in vivo proliferation been tested in a separate cohort of animal tumors to establish whether MK inhibits proliferation in vivo?
3. It appears from the in vitro results that CXCL1 is a key mediator of MK's effects on proliferation. It would be very informative if anti-CXCL1 antibody administered in vivo could recapitulate some of the tumor-inhibitory effects of the MKC inhibitor in combination with paclitaxel; though it is recognized that such experiment could be costly.
4. What is the mechanism behind inhibition of the cytokine expression inhibition by MKC8866? Since it appears that the effect is at the transcriptional level, have the authors analyzed the effects of XBP1 inhibition on NF-kappaB or other transcription factors which could induce multi-cytokine activation?
5. Some of the results are presented out of order, which can be rather confusing. For example, the authors report the RNAseq data from Control and MKC8866-treated cells without going through the results to focus on the secretome, and present it in more detail later. Consider reversing the order and reporting the complete RNAseq data first.

6. The IHC results shown in Fig. 4k are of poor quality. While the XBP1 staining is clear, that of CXCL1 and IL-8 is not. Moreover, there is no mention of how the results were quantified and whether an independent, blinded investigator determined staining intensity or area of expression.

Minor point:

a. At the bottom of p.5, the authors state that "...MKCC6688 blocked Tm-induced IRE-1 mediated signaling but did not affect.....ATF6 signaling". ATF6 signaling has not been tested, recommend changing to "ATF6 processing".

Constantinos Koumenis

Response to Reviewers' comments:

Reviewer #1 (Remarks to the Author):

In this manuscript, Logue et al. examine the role of IRE1 α functionally in a number of triple negative breast cancer (TNBC) cells lines and using biomarkers in human tumor samples. As has been previously reported by others (Chen X et al. Nature 2014; 508:103-7), they find evidence that the IRE1/XBP1 pathway is activated in triple negative breast cancer cell lines and human samples. Moreover, they show that knocking down XBP1 or using a small molecule inhibitor of IRE1 (MKC8866) has anti-proliferative effects on TNBC cell lines grown in culture and is associated with a reduction in the production of a number of pro-inflammatory cytokines (such as CXCL-1 and IL-8). Using xenograft studies in mice, they find that monotherapy with MKC8866 has no significant effects on tumor growth. However, they show that treating cultured TNBC cell lines with paclitaxel upregulates XBP1 and some of the before mentioned cytokines. They then see that the combination of MKC8866 plus paclitaxel results in modestly better anti-tumor growth effects and survival benefits in the xenograft model than paclitaxel alone.

Overall, the findings are interesting and may provide further evidence that IRE1 is a potential target in TNBC, a malignancy for which much better therapies are needed. However, in its present form, the manuscript has a number of major flaws and missing controls that must be addressed before any conclusions can be drawn about the role of IRE1 in this model of TNBC.

Major Concerns:

1) Many of the studies here rely heavily on the use of MKC8866, an aldehyde-based inhibitor that forms a Schiff base with a lysine (K907) in the RNase active site of IRE1 α . In the abstract, the authors inaccurately describe MKC8866 as “novel small molecule” when it was originally reported in 2011 (Volkman K et al. J Biol Chem 286; 12743-12755). Hence, it is hardly novel and that term needs to be removed from the manuscript.

Authors' Response: We thank the reviewer for their feedback and as requested we have removed the term novel from the manuscript.

Second, the authors state in the abstract that this molecule is a “highly selective IRE1 RNase inhibitor.” While it is clear that this compound inhibits IRE1 α 's RNase activity in cell culture when used at high concentrations (25 μ M), what is the evidence for selectivity of this compound (showing that PERK and ATF6 are not block is hardly evidence of monoselectivity)? How many other lysines on others proteins do they bind? We have no idea and therefore need to be very cautious with interpreting any results with these compounds (especially as they result to cytotoxicity).

Authors' response: We agree with the reviewer that the specificity and selectivity of the inhibitor is very important for the correct interpretation of the findings. For that reason we have also used genetic approaches such as siRNA knockdown to validate results achieved using MKC8866.

While we cannot absolutely exclude the possibility of MKC8866 binding to lysines present on other proteins, publications by Volkmann et al. (Volkmann *et al.*, 2011, J Biol Chem, 286: 12743-12755), and Sanches et al. (Sanches M *et al.*, 2014, Nat Comms, 5, 4202) suggest the interaction between hydroxyl-aryl-aldehyde inhibitors and the IRE1 RNase domain is highly specific and selective. The RNase domain of IRE1alpha is very unusual and is found in only one other paralogue, RNase L. Hydroxyl-aryl-aldehydes, including MKC8866, while inhibiting IRE1 RNase activity, failed to block activity of RNase L due to the absence of a suitably positioned lysine residue for Schiff base formation. Recent studies by Frank Sicheri's group have demonstrated that in addition to having a suitably positioned lysine residue for Schiff base formation, the structure of the binding pocket is important in determining the binding affinity of hydroxyl-aryl-aldehyde inhibitors. Sanches et al. demonstrated MKC9989 (closely related to MKC8866) requires several other amino acid residues within the binding pocket to facilitate formation of a reversible Schiff base interaction between the aldehyde moiety of MKC9989 and Lys907 in the RNase (Sanches M *et al.*, 2014, Nat Comms, 5, 4202). This indicates that, in addition to having a suitably positioned lysine residue, the composition/configuration of the binding pocket is important in determining the selectivity of hydroxyl-aryl-aldehydes inhibitors. To definitively establish the global selectivity profile of MKC8866 a broad chemical proteomics study would be required, which is beyond the scope of this manuscript.

While not diminishing the reviewer's valid concerns we hope our additional *in vitro* experiments, including genetic studies will help to reduce concerns regarding inhibitor specificity in our study. In addition we have changed the statement "highly selective IRE1 RNase inhibitor" to "selective IRE1 RNase inhibitor in consideration of the reviewer's comment.

2) Given the questionable selectivity of MKC8866, I very much appreciate the authors' efforts to use shRNA knockdown as a second genetic approach. However, a number of additional controls need to be included. In addition to knocking down XBP1 in MDA-MB-231 cells in Figure 2, they also need to independently knockdown IRE1. Moreover, they then need to test the effect of MKC8866 on the cells knocked down for IRE1 and XBP1. Moreover, they need to also test the effects of MKC8866 on the MDA-MB-231 cells where they over-express XBP1s (these cells should be highly resistant to any effects of this compound).

Authors' response: We thank the reviewer for these comments and have now carried out these experiments with the additional controls requested by the reviewers.

To address the first part of this comment we examined the effect of MKC8866 on MDA-MB-231 cells stably expressing XBP1-shRNA or with transient knockdown of IRE1. Both of these knockdown

cell lines exhibited a decrease in proliferation compared with control cells. Addition of MKC8866 reduced cell proliferation in the control cells as expected (without inducing cell death), and to a lesser extent in cells where XBP1 or IRE1 was knocked down. These are presented in **Supplemental Figure 1I-K**. Since these cells were knockdown rather than knockout it was not surprising that MKC8866 reduced their proliferation to some extent. These data implicate IRE1 signaling in the maintenance of cell proliferation and also support the selectivity of MKC8866 for IRE1 RNase activity. Since the initial submission of this manuscript two other groups have linked IRE1 signaling to cell proliferation, where knockdown of IRE1 slowed the growth of colon cancer cells (Li *et al.* *Oncogene*. 2017, 36(48):6738-6746), and both MKC8866 and another inhibitor of IRE1 RNase activity were shown to reduce proliferation of mast cell leukemia cell lines (Wilhelm T, *et al.* *Oncotarget*. 2017, 9(3):2984-3000).

To address the second part of this comment we transiently overexpressed XBP1s in MDA-MB-231 cells. Compared with empty vector (EV) controls, XBP1s overexpression significantly increased cell proliferation. Addition of MKC8866, as expected, reduced cell proliferation in EV-expressing cells. Unexpectedly, a similar proportional reduction in cell number was also observed in XBP1s-overexpressing cells treated with MKC8866. To understand the reason for this significant reduction, as suggested by the reviewer (point 3) we assessed XBP1s levels during the course of the experiment. While the cells overexpressed XBP1s at day 0, this overexpression was no longer evident by day 6 (**Figure A**). This loss in XBP1s overexpression may explain the reduction in proliferation we observed upon MKC8866 addition to these cells. Our interpretation of these results is that overexpression of XBP1s, although transient, boosted cell proliferation at earlier time points in the 6-day experiment. Given the lack of sustained XBP1s overexpression we have removed figure 2K-L from the manuscript.

Figure A

MDA-MB-231 cells were transfected with pCNDNA3.1 or pcDNA3.1-hXBP1s. 20 h post-transfection cells were reseeded and treated with vehicle alone or 20 μ M MKC8866. On day 2 post-treatment, cells were again transfected with pCNDNA3.1 or pcDNA3.1-hXBP1s, reseeded and treated with vehicle or MKC8866 every second day until day 6. Cell lysates were collected every second day and immunoblotted for XBP1s (upper band). Cell proliferation was monitored by cell counts. Counts shown are for day 6 \pm SEM.

3) In Figure 2K-L, they need to perform an immunoblot for XBP1s not only after 24 hrs but also in these cells for the duration of the experiment (8 days). We and others have found that many cell types do not allow XBP1s to continue to be over-expressed as a transgene after 24-48hrs through mechanisms that are unclear.

Authors' response: As described in our response to point 2 above, we assessed XBP1s levels during the course of the experiment and while the cells overexpressed XBP1s at day 0, the overexpression was lost by day 6 (Figure A). Based on these findings we have chosen to remove Figure 2K-L from the manuscript.

4) In Figure 3, they need to test the effects of over-expressing and blocking the relevant cytokines (CXCL1) on MDA-MB-231 proliferation as assayed in Figure 2.

Authors' response: We thank the reviewer for this excellent suggestion. We observed that addition of neutralizing antibodies against cytokines regulated in an IRE1-dependent manner, including CXCL1, reduced MDA-MB-231 cell proliferation (as determined by a decrease in cell number that was not associated with an increase in cell death) to a level that was similar to that observed upon MKC8866 addition. This suggests cytokines, presumably via autocrine signaling, contribute in part to cell proliferation (Supplemental Figure 3I).

As suggested by the reviewer, we also assessed if addition of exogenous recombinant CXCL1, was able to overcome the suppressive effect of MKC8866 on proliferation. As shown in Figure B, addition of recombinant CXCL1 did not alter MKC-mediated suppression of proliferation. While our data demonstrates IRE1-regulated cytokines can contribute to cell proliferation, a direct link between MKC8866-mediated inhibition of proliferation and reduced CXCL1 production was inconclusive.

Figure B

MDA-MB-231 cells were treated for 5 days with vehicle alone (DMSO), MKC8866 alone (20 μ M), or a combination of MKC8866 and recombinant CXCL1 (500 pg/ml). MKC8866 and recombinant CXCL1 were replenished every 2 days. Cell number was determined by cell counts on day 5.

5) In Figure 4, the IHC images for XBP1s are of such low magnification and quality, that it is impossible to determine if the staining is nuclear or non-specific. They need to provide much better and higher quality images or remove these studies from the manuscript.

Authors' response: We have included a high magnification (40X) image of XBP1s staining below in **Figure C**. XBP1s specific staining appears to be largely nuclear. Unfortunately we were unable generate high magnification images for IL8 and CXCL1 of sufficient quality for publication (as requested by reviewer 2) and so have chosen to remove the panel from the manuscript.

Figure C

Representative example of XBP1s staining at 40X magnification in TNBC tissue array.

6) In Figure 6, they need to test the effects of the neutralizing antibodies on their own (in the absence of paclitaxel).

Authors' response: We appreciate the reviewer's comment and have now performed this experiment. As shown in **Supplemental Figure 3L** the addition of the neutralizing antibodies alone (in the absence of paclitaxel) does not alter the ability of MDA-MB-231 cells to form mammospheres following 5-day culture on ultra-low adherence plates.

7) The lack of effect of MKC8866 monotherapy in the xenograft studies is disappointing and challenges the central hypothesis here. They need to harvest the tumors from these MKC8866 treated animals and confirm that IRE1 is indeed inhibited *in vivo* (XBP1 splicing). Moreover, given the concerns raised above regarding the selectivity of MKC8866, they should also genetically knockdown IRE1 and XBP1 in this cell line prior to injecting it as a xenograft as a second approach.

Authors' response: As the reviewer rightly points out it is important to verify the on-target effect of MKC8866 *in vivo*. Using qPCR we have demonstrated reduced XBP1 splicing in tumors from mice receiving MKC8866 compared to vehicle alone confirming a reduction in IRE1 RNase activity *in vivo* (**Figure 7B**). Additionally we also examined XBP1 splicing in tumors from vehicle-only, paclitaxel-only and paclitaxel plus MKC8866 treatment groups. As shown in **Figure 7D**, paclitaxel led to increased XBP1 splicing, which was reduced by combination with MKC8866.

To answer the second part of this point, like the reviewer, we originally considered generating IRE1 or XBP1 knockout MDA-MB-231 cell lines to use in the *in vivo* studies. However, upon careful consideration we decided against this approach for several reasons. Primarily, we were concerned that cells deficient in IRE1 or XBP1 would not be able to form tumors *in vivo* as efficiently as control cells. It has been previously reported that XBP1 is required for tumor growth, and that XBP1^{-/-} MEF cells fail to form tumors in contrast to XBP1^{+/+} MEF cells (Romero-Ramirez L, *et al.* Cancer Res. 2004, 64(17):5943-7). Therefore, the question that would be addressed in such an experiment switches from assessing the potential of IRE1 as therapeutic target in pre-existing tumors to asking the importance of IRE1-XBP1 signaling in xenograft development, and thus the result of such an experiment could not be interpreted easily. Taking into account the reasons outlined above, along with financial and ethical considerations, we therefore decided against these studies.

8) The results in Figure 8 are not at all convincing. Is there a significant difference between the MKC8866 + paclitaxel treated animals vs the paclitaxel alone animals (what is the p-value)? If not, this figure should be removed.

Authors' response: When checking the data for this figure we realized that we had mistakenly plotted the data points for 150 mg/kg MKC8866 rather than the 300 mg/kg as indicated in the figure legend. We have now plotted vehicle versus 300 mg/kg MKC8866 and indicated the time points at which the difference in tumor volume between the two groups is statistically significant. There is a statistically significant difference ($p < 0.05$) between the 'MKC+paclitaxel' and 'paclitaxel alone' groups at days 22, 25, 29, 32 and 36. After this point the tumor volume in the 'MKC+paclitaxel' group was smaller than the 'paclitaxel alone' group, but the difference was not statistically significant. These data suggest that maintaining suppression of IRE1 RNase signaling via MKC8866 administration could hinder tumor regrowth post-paclitaxel treatment.

Reviewer #2 (Remarks to the Author):

The manuscript by Logue and colleagues presents some novel and clinically relevant results linking inhibition of the RNase activity of the UPR mediator IRE1 to proliferation arrest and *in vivo* growth of triple-negative breast cancer (TNBC) cell lines treated with paclitaxel. The link between TNBC and IRE1/XBP-1 activation has been firmly established (Chen et al, Nature, 2014). What is novel in this manuscript is the demonstration that inhibition of IRE1's RNase activity significantly attenuates autocrine and paracrine signaling of pro-tumorigenic cytokines and synergizes with paclitaxel to confer potent anti-tumor effects. Moreover, the finding that TNBC with a higher IRE1 RNase signature, especially with elevated IL-8 and CXCL1 levels are associated with more advanced clinical grade is novel and clinically important. The manuscript is well-written (although some reorganization of sections is recommended) and the data is clear. There are some concerns however regarding the specific mechanism behind the observed anti-cytokine effects and the interpretation of the anti-tumor effects of the MKC8866 inhibitor in combination with paclitaxel.

Major comments:

1. Since a previous publication has linked XBP1 activation in TNBC to regulation of HIF1, it is possible that *in vivo* some or all the observed growth-inhibitory effects are due to defective angiogenesis. This is also likely because some of the cytokines observed to be affected by the MK inhibitor are potential HIF1 targets (Chen et al, 2014). Analysis of angiogenesis in tumors (e.g., by analyzing microvessel density in tumors), as well as HIF1 levels needs to be analyzed.

Authors' response: We agree that this is a possibility. Indeed, while we were revising this paper, another paper (Zhao *et al.*, JCI 2018, Feb 26. pii: 95873. doi: 10.1172/JCI95873), demonstrated that MKC8866 can decrease CD31⁺ levels, indicating that it can reduce blood vessel formation *in vivo*. Additionally, Lhomond *et al.* have recently reported increased angiogenesis in glioblastoma upon elevated IRE1-XBP1s signaling (Lhomond et al. EMBO Mol Med 2018, Jan 8. pii: e7929). We have included references to both of these papers in our discussion. Owing to time constraints we have not been able to assess microvessel density or HIF1 expression in our tumor sections via IHC. To address the reviewers comment, we assessed transcript levels of pro-angiogenic factors *VEGF* and *IL8* in tumors following treatment with paclitaxel or paclitaxel plus MKC8866. As presented below in **Figure D** combination with MKC8866 reduced transcript levels of *IL8* and *XBP1s* but not *VEGFA* in paclitaxel treated tumors. Further studies are needed to definitively assess the outcome of MKC8866 on angiogenesis

Figure D

Assessment of *XBPIs*, *IL8* and *VEGFA* transcript levels in paclitaxel and paclitaxel plus MKC8866 treated tumor xenografts. N=4 in each treatment group.

2. The *in vitro* results clearly establish that the MKC8866 inhibitor attenuates TNBC proliferation alone. Yet, *in vivo*, the MKC8866 drug has no effect, but does synergize with paclitaxel to inhibit tumor growth. Why the discrepancy? Has *in vivo* proliferation been tested in a separate cohort of animal tumors to establish whether MK inhibits proliferation *in vivo*?

Authors' response: The reviewer raises a very interesting point. We believe the discrepancy observed in terms of proliferation *in vitro* and *in vivo* is due to the manner in which the *in vivo* xenograft experiment was conducted compared with the *in vitro* experiments. *In vivo*, the tumors were allowed to grow to 250 mm³ before addition of the IRE1 inhibitor, by which stage the size of the tumor may hamper detection of differences due to the inhibitor. Moreover, addition of paclitaxel to these tumors further enhances IRE1 activity and may increase tumor cell dependence on IRE1 activity. Therefore, blocking IRE1 activity in this setting may have a more significant outcome resulting in reduced tumor volume. In contrast, the *in vitro* studies were performed with cells in the log phase of growth and low seeding densities.

We have not performed *in vivo* proliferation experiments to determine whether MKC inhibits proliferation *in vivo*, mainly owing to constraints of cost and time. However, based on previously published work (Romero-Ramirez L, *et al.* Cancer Res. 2004 Sep 1;64(17):5943-7) we would expect MKC8866 to slow tumor formation *in vivo*.

3. It appears from the *in vitro* results that CXCL1 is a key mediator of MK's effects on proliferation. It would be very informative if anti-CXCL1 antibody administered *in vivo* could recapitulate some of the tumor-inhibitory effects of the MKC inhibitor in combination with paclitaxel; though it is recognized that such experiment could be costly.

Authors' response: This would appear to be a logical experiment to do; however it would be highly complex and, as the reviewer points out, the cost of such an experiment is prohibitive. The experiment would require extensive optimization to determine, for example, what concentration of CXCL1 neutralizing antibody to use, how to administer (systemically or direct to the tumor), how much actually reaches the tumor cells and finally the frequency of administration. However, in response to a suggestion by Reviewer 1 we examined the effect of CXCL1 neutralizing antibodies on cell proliferation, and observed a decrease. We have also strengthened the data that links CXCL1 to the paclitaxel driven increase in tumor initiating cells. In addition to demonstrating that CXCL1 neutralization reduces mammosphere formation, we also showed that addition of recombinant CXCL1 partially reverses MKC-mediated suppression of mammosphere formation (**Figure 6D**).

4. What is the mechanism behind inhibition of the cytokine expression inhibition by MKC8866? Since it appears that the effect is at the transcriptional level, have the authors analyzed the effects of XBP1 inhibition on NF-kappaB or other transcription factors which could induce multi-cytokine activation?

Authors' response: We agree with the reviewer that this is an important question. To address this, we examined the effect of MKC8866 treatment on a panel of potential transcription factors including NF-kB, HIF1 α and STAT3. In addition we also examined β -catenin and FOXO1 expression as both transcription factors have recently been reported to be regulated by IRE1 signaling and can contribute, although indirectly, to cytokine production. This showed that MDA-MB-231 cells have constitutively active NF-kB (as demonstrated by constitutive phosphorylation of p65) and constitutive STAT3 phosphorylation. However, treatment with MKC8866 did not alter the phosphorylation status of p65 or STAT3, indicating that the reduction in cytokine production is not dependent on modulation of NF-kB or STAT3 activity. Likewise, we observed no changes in β -catenin, FOXO1 or HIF1 α expression following treatment with MKC8866 (**Supplemental Figure 5A**).

While this experiment does not reveal a definitive mechanism, it does rule out some likely transcription factors, including those mentioned by the reviewer.

5. Some of the results are presented out of order, which can be rather confusing. For example, the authors report the RNAseq data from Control and MKC8866-treated cells without going through the results to focus on the secretome, and present it in more detail later. Consider reversing the order and reporting the complete RNAseq data first.

Authors' response: We thank the reviewer for this comment. We agree with the suggestion and have now reordered the results accordingly. We have also added in **Supplemental Figure 2** showing GO enrichment analysis of the transcriptome data, which initially flagged a link between IRE1 RNase activity and inflammation and which subsequently prompted us to investigate the secretome.

6. The IHC results shown in Fig. 4k are of poor quality. While the XBP1 staining is clear, that of CXCL1 and IL-8 is not. Moreover, there is no mention of how the results were quantified and whether an independent, blinded investigator determined staining intensity or area of expression.

Authors' response: The following details have now been added to the materials and methods section detailing how IHC was carried out "Immunohistochemistry staining was quantified by an independent investigator using a blinded approach according to the methodology outlined by Varghese et al. (Varghese *et al.* 2014. Plos One 9(5) e96801). Unfortunately, we were unable generate high

magnification images for IL-8 and CXCL1 of sufficient quality for publication and so have chosen to remove panel from the manuscript

Minor point:

a. At the bottom of p.5, the authors state that “...MKCC6688 blocked Tm-induced IRE-1 mediated signaling but did not affect....ATF6 signaling”. ATF6 signaling has not been tested, recommend changing to “ATF6 processing”.

Authors' response: We thank the reviewer for drawing our attention to this error. We have corrected it in the revised manuscript.

Reviewers' comments:

Reviewer #2 (Remarks to the Author):

I appreciate the authors efforts to address my concerns with additional experiments. They have improved certain things (such as the immunohistochemistry for XBP1) and removed the XBP1s over-expression data.

However, after reviewing the latest data I remain unconvinced about several things.

1) Based on the sum of their data, I remain skeptical that the anti-proliferative effects seen with MKC8866 are largely the result of IRE1 inhibition, and hence should be presented more objectively. Their explanations on the previously published data for this compound are not helpful. For example, the observation that MKC8866 does not inhibit IRE1's closest paralog RNase L is non-contributory since the later does not contain a similarly positioned lysine with which to form a Schiff base. Given the mechanism of action, the best way to look for off target effects is to remove IRE1 from cells and see if MKC8866 still reducing proliferation. When the authors performed this experiment MDA- MB-231 cells stably expressing XBP1-shRNA or with transient knockdown of IRE1, addition of MKC8866 still reduced cell proliferation (albeit to a lesser extent than wild-type cells). The authors try to argue this is due to partial knockdown of the targets. To test this idea, they should employ CRISPR/Cas9 to completely eliminate expression of IRE1 and XBP1 and test the effects of the drug.

To my eyes, their results suggest that the MKC8866 anti-proliferative effects are partly through IRE1 and partly through off-target effects.

2) The new data using CXCL1 antibodies do not support the model that this is a critical cytokine downstream of IRE1, which when reduced by IRE1 inhibition causes the cells to stop proliferating. Recombinant CXCL1 did not alter MKC-mediated suppression of proliferation (however, given my prior concerns about MCK8866, it would be better to do this experiment on IRE1 knockout cells). Hence, they don't have a solid mechanistic explanation for how IRE1 inhibition blocks proliferation.

3) The lack of effect of MKC8866 monotherapy in the xenograft studies continues to challenge the central hypothesis here. To their credit, they have now harvested the tumors from these MKC8866 treated animals and confirmed that IRE1 is indeed inhibited in vivo (XBP1 splicing). However, they argue against doing IRE1 or XBP1 knockdown studies in vivo, which are necessary especially given the concerns about that selectivity of MCK8866. They should use an inducible knockdown or knockout approach if they are worried about untangled the effects on tumor development vs the growth of a preformed tumor.

Response to Reviewers' comments:

1) Their explanations on the previously published data for this compound are not helpful. For example, the observation that MKC8866 does not inhibit IRE1's closest paralog RNase L is non-contributory since the later does not contain a similarly positioned lysine with which to form a Schiff base.

In the original feedback Reviewer 1 stated “**How many other lysines on others proteins do they bind? We have no idea and therefore need to be very cautious with interpreting any results with these compounds (especially as they result to cytotoxicity)**”. In our rebuttal we provided a detailed overview, based primarily on the 2014 Nature Communications article by Sanches et al which solved the crystal structure of murine IRE1alpha in complex with three hydroxy-aryl-aldehydes inhibitors including MKC9989 (closely related to MKC8866). This study provides a detailed analysis of inhibitor binding and demonstrates that in addition to having a suitably positioned lysine residue for Schiff base formation, the structure of the binding pocket is important in determining the binding affinity of hydroxyl-aryl-aldehyde inhibitors. We do not understand why a reviewer would dismiss the peer-reviewed published literature describing it as “not helpful”. This inhibitor does not inhibit any other RNase, hence the reference to RNase L. While, we cannot discount binding of the drug to another target, there is no evidence that it does bind to any other protein.

Additionally the reviewer states these compounds (presumably the MKC inhibitor family) result in cytotoxicity but does not provide a reference to qualify this statement. As demonstrated in Supplementary Figure 1A-C we observed no evidence of cytotoxicity in three breast cancer cell lines (MCF7, MDA-MB-231 and SKBR3) after 6 days of culture with 20 μ M MKC8866. However reduced proliferation following MKC8866 treatment was evident in each of these cell lines (Figure 2E-G). A study recently published by Wilhelm and colleagues examining the outcome of MKC8866 treatment on the proliferation of HMC-1.2 cells reported 20 μ M MKC8866 reduced cell proliferation but did not induce cell death (Wilhelm T, *et al.* Oncotarget. 2017, 9(3):2984-3000, Figure 1).

Based on our findings and the published literature on MKC8866 there is no evidence to support the statement by the reviewer that MKC8866 is cytotoxic when used at 20 μ M or up to 300 mg/kg in vivo. We fail to understand on what basis the reviewer reached this conclusion and can only conclude that this is a personal opinion of the reviewer.

Given the mechanism of action, the best way to look for off target effects is to remove IRE1 from cells and see if MKC8866 still reducing proliferation. When the authors performed this experiment MDA-MB-231 cells stably expressing XBP1-shRNA or with transient knockdown of IRE1, addition of MKC8866 still reduced cell proliferation (albeit to a lesser extent than wild-type cells). The authors try to argue this is due to partial knockdown of the targets. To test this idea, they should employ CRISPR/Cas9 to completely eliminate expression of IRE1 and XBP1 and test the effects of the drug.

In the original submission of the manuscript we provided evidence of reduced proliferation by MKC8866 of BC cell lines that express basal IRE1 activity and XBP1 splicing, which was not observed in a BC cell line (the non-cancerous cell line MCF10a) that lacks basal IRE1 activity and XBP1 splicing.

We were originally asked by Reviewer 1 to perform additional experiments adding MKC8866 to both empty vector and XBP1shRNA in MDA-MB-231 cells and to examine the effect on proliferation. As shown in Supplemental Figure 1I, we observed a substantial decrease (73%, $p < 0.001$) in cell proliferation in empty vector cells following treatment with MKC8866 for 5 days. We also found that MKC8866 reduced proliferation in XBP1shRNA cells but to a much lesser extent (36%, $p < 0.05$). Indeed there was no significant difference between the cell number following MKC8866 treatment in XBP1shRNA cells and the cell number in empty vector cells post MKC8866 treatment. However, as shown in Figure 2I, XBP1s induction following treatment with thapsigargin was still evident in XBP1shRNA cells due to non-complete silencing of XBP1s with the shRNA strategy. Therefore, we concluded that the reduction observed in MKC8866-treated XBP1shRNA cells was due to residual XBP1s expression as a consequence of the cells being knockdown and not knockout.

As also requested by the Reviewer we showed that siRNA-mediated IRE1 silencing in MDA-MB-231 cells decreased cell proliferation compared to non-coding control siRNA transfected cells (Supplementary Figure 1J-K). Similar to XBP1 knockdown cells, we found that while addition of MKC8866 substantially reduced proliferation in control siRNA cells it did not significantly alter cell number in IRE1 knockdown cells.

Our results show that the IRE1-XBP1 signaling axis is important in maintaining cell proliferation in breast cancer cells and that the constitutive activation of this signaling axis renders cells sensitive to MKC8866. This observation also agrees with recently published literature (Li *et al.* *Oncogene*. 2017, 36(48):6738-6746; Wilhelm T, *et al.* *Oncotarget*. 2017, 9(3):2984-3000). While the reviewer does not question the fact that IRE1-XBP1 axis contributes to cell proliferation, he/she suggests that MKC8866 may be further reducing cell proliferation through off-target effects and requested that we generate IRE1 and XBP1 CRISPR cell lines to test selectivity and specificity of MKC8866. We have now knocked out XBP1 (using CRISPER/Cas9) and include the data in the figure below. XBP1 knockout reduced cell proliferation in 2 clones. Furthermore, although MKC8866 reduced proliferation of the control cells, it did not impact on proliferation of XBP1 knockout cell. If MKC8866 had significant off-target effects it should be evident in MCF10a cells. We feel that collectively these data demonstrate MKC8866's specificity as far as can reasonably be expected.

XBP1 knockout abolishes effect of MKC8866 (A) Proliferation of control (scrambled guide RNA transfected) and XBP1 knockout MDA-MB-231 cell lines was followed over a 7 day period in the presence of vehicle only (DMSO) or MKC8866 (20 μ M). (B) MKC8866-mediated suppression of cell proliferation at Day 7 in control (scrambled guide RNA transfected) and XBP1 knockout MDA-MB-231 cell lines.

Our study uses MKC8866 as a tool, in combination with genetic approaches (which yield convergent results with those obtained with MKC8866), to address the role of IRE1 signaling in triple negative breast cancer. We feel that our genetic studies have demonstrated that IRE1-XBP1 signaling contributes to the proliferative capacity of breast cancer cell lines. We do not think that it is fair or reasonable of this reviewer to now suggest additional time consuming experiments designed not to advance the scientific findings of our study but only to test MKC8866 selectivity.

To my eyes, their results suggest that the MKC8866 anti-proliferative effects are partly through IRE1 and partly through off-target effects.

We show that addition of MKC8866 to MCF10a cells had no effect on cell proliferation. If MKC8866 had off-target effects it should be evident in any cell line and therefore systematically exhibit anti-proliferative effects, which is not the case. Also, the XBP1 knockout experiment presented above supports an effect mediated by IRE1-XBP1 signaling.

2) The new data using CXCL1 antibodies do not support the model that this is a critical cytokine downstream of IRE1, which when reduced by IRE1 inhibition causes the cells to stop proliferating. Recombinant CXCL1 did not alter MKC-mediated suppression of proliferation (however, given my prior concerns about MCK8866, it would be better to do this experiment on IRE1 knockout cells). Hence, they don't have a solid mechanistic explanation for how IRE1 inhibition blocks proliferation.

Neither in the original nor in the revised manuscript did we state that CXCL1 is a critical cytokine downstream of IRE1 driving cell proliferation. We have linked CXCL1 increases post-paclitaxel treatment to the expansion of tumor initiating cells. We demonstrated that neutralization of CXCL1 reduced tumor initiating cell expansion (based on mammosphere formation) and that addition of

recombinant CXCL1 was able to overcome MKC-mediated suppression of tumor initiating cell expansion post-paclitaxel treatment.

The experiments with neutralizing antibodies originally requested by the reviewer demonstrate a role for cytokines in the proliferation of MDA-MB-231 cells. However, we think it is unlikely that regulation of cytokines is the only factor through which IRE1 signaling contributes to cell proliferation. For this reason we did not conclude in the manuscript that CXCL1 regulation was a regulator of cell proliferation, rather than it is a contributor to expansion of tumor initiating cells.

3) The lack of effect of MKC8866 monotherapy in the xenograft studies continues to challenge the central hypothesis here. To their credit, they have now harvested the tumors from these MKC8866 treated animals and confirmed that IRE1 is indeed inhibited in vivo (XBP1 splicing). However, they argue against doing IRE1 or XBP1 knockdown studies in vivo, which are necessary especially given the concerns about that selectivity of MCK8866. They should use an inducible knockdown or knockout approach if they are worried about untangled the effects on tumor development vs the growth of a preformed tumor.

As requested by Reviewer 1 we have provided evidence verifying on-target effect of MKC8866 in MDA-MB-231 xenografts treated with inhibitor alone or a combination of MKC8866 and paclitaxel. The reviewer now suggests that we generate inducible IRE1/XBP1 knockdown or knockout MDA-MB-231 cells, establish xenografts with these cells, and induce IRE1/XBP1 knockdown/knockout to test the specificity of MKC8866. We find this suggestion surprising and unnecessary when none of our data suggests in any way that MKC8866 has significant off-target effects. Two additional studies utilizing MKC8866 in vivo have recently been published (Zhao et al, 2018, J Clin Invest, doi:10.1172/JCI95873; Rubio-Patiño C et al, 2018, Cell Metabolism, doi: 0.1016/j.cmet.2018.02.009). Like our study, both of these recent publications demonstrated MKC8866 efficacy by analyzing XBP1 splicing. The publication of Zhao et al also investigated the outcome of long-term systemic administration of MKC8866 on mice and observed no evidence of toxicity or off-target effects.

As the reviewer states our results show that as a single agent MKC8866, while blocking IRE1 signaling, did not reduce tumor growth. However, the combination of MKC8866 and paclitaxel caused a greater reduction in tumor volume compared with paclitaxel alone (Figure 7C). Treatment with paclitaxel produced an increase in IRE1 RNase activity in the tumor, as shown by increased XBP1 splicing which is inhibited by treatment of MKC8866 (Figure 7D). We argue that this increase in XBP1s is reflective of elevated UPR within the tumor and that blocking IRE1-XBP1 signaling axis tips the balance within cells in the direction of cell death. We suspect that the lack of effect on tumor growth as a single agent may be a reflection of the genetic makeup of the tumor, in particular the levels of MYC. While we found MKC8866 did not impact on the growth of MDA-MB-231 xenografts, Zhao and colleagues recently reported that MKC8866 only works as a single agent in those xenografts derived from cells displaying a particularly high expression of MYC.

REVIEWERS' COMMENTS:

Reviewer #2 (Remarks to the Author):

Through a series of additional experiments, the authors have now addressed all my major concerns. As a result, the manuscript is much stronger and now acceptable for publication.